# The L-NAME mouse model of preeclampsia and impact to long-term maternal cardiovascular health

Natasha de Alwis[1,2] , Natalie K Binder[1,2], Sally Beard[1,2], Yeukai TM Mangwiro[1,2], Elif Kadife[2,3], James SM Cuffe[4] , Emerson Keenan[2,3] , Bianca R Fato[1,2], Tu'uhevaha J Kaitu'u-Lino[2,5], Fiona C Brownfoot[2,3], Sarah A Marshall[6] , Natalie J Hannan[1,2]

**Preeclampsia affects ~2–8% of pregnancies worldwide. It is associated with increased long-term maternal cardiovascular disease risk. This study assesses the effect of the vasoconstrictor N(ω)-nitro-L-arginine methyl ester (L-NAME) in modelling preeclampsia in mice, and its long-term effects on maternal cardiovascular health. In this study, we found that L-NAME administration mimicked key characteristics of preeclampsia, including elevated blood pressure, impaired fetal and placental growth, and increased circulating endothelin-1 (vasoconstrictor), soluble fms-like tyrosine kinase-1 (anti-angiogenic factor), and C-reactive protein (inflammatory marker). Post-delivery, mice that received L-NAME in pregnancy recovered, with no discernible changes in measured cardiovascular indices at 1-, 2-, and 4-wk post-delivery, compared with matched controls. At 10-wk post-delivery, arteries collected from the L-NAME mice constricted significantly more to phenylephrine than controls. In addition, these mice had increased kidney *Mmp9:Timp1* and heart *Tnf* mRNA expression, indicating increased inflammation. These findings suggest that though administration of L-NAME in mice certainly models key characteristics of preeclampsia during pregnancy, it does not appear to model the adverse increase in cardiovascular disease risk seen in individuals after preeclampsia.**

## Introduction

Preeclampsia is a severe obstetric disease affecting ~2–8% of pregnancies worldwide (1, 2, 3). Its pathogenesis stems from the dysfunctional placenta, which releases anti-angiogenic and pro-inflammatory factors into the maternal circulation, resulting in widespread endothelial dysfunction (4, 5, 6). Endothelial dysfunction

is responsible for systemic vasoconstriction and chronic hypertension, limiting blood supply to major organs (7). Consequently, preeclampsia can cause major organ injury and impair fetal development, making it responsible for significant maternal and perinatal morbidity and mortality (8). In addition, preeclampsia also impacts long-term health of both the pregnant individual and child (9). It is well established that individuals whose pregnancies were complicated by preeclampsia or gestational hypertension are at higher risk of cardiovascular disease post-delivery than those with normotensive pregnancies (10, 11, 12, 13, 14).

There is currently no cure for preeclampsia, and limited treatment options. Hence, there is urgent need for development of therapeutics that directly target the disease. Animal models play an important role in understanding the pathophysiology that drives preeclampsia, and offer valuable potential in the development and testing of candidate therapeutics in an in vivo system. Established preeclampsia models have been summarised in several studies (15, 16, 17, 18), but it is well accepted that one model alone is unlikely to encompass all aspects of preeclampsia pathogenesis due to the complexity and heterogeneity of the disease. Rather, each model might mimic certain key pathways or aspects of the disease process. Henceforth, assessing the benefits of a candidate therapeutic may be best performed through multiple models examining the effects over several key pathways that underpin preeclampsia.

However, most existing models of preeclampsia do not encompass the long-term effects of the disease (19, 20, 21, 22). This is a gap in the knowledge and tools available for enhancing preeclampsia research. A part of assessment of preeclampsia therapeutics should be to examine whether they can also improve long-term outcomes. This is particularly important for long-term maternal health, as cardiovascular disease is a leading cause of death of women in Australia (23). Pregnancy provides a unique window for intervention, given pregnancy complications can impact long-term maternal health and increase risk of cardiovascular disease. Animal models of preeclampsia with

[1]Department of Obstetrics and Gynaecology, Therapeutics Discovery and Vascular Function Group, The University of Melbourne and Mercy Hospital for Women, Heidelberg, Australia [2]Mercy Perinatal, Heidelberg, Australia [3]Department of Obstetrics and Gynaecology, Obstetrics Diagnostics and Therapeutics Group, The University of Melbourne and Mercy Hospital for Women, Heidelberg, Australia [4]School of Biomedical Sciences, The University of Queensland, Brisbane, Australia [5]Department of Obstetrics and Gynaecology, Diagnostics Discovery and Reverse Translation in Pregnancy Group, The University of Melbourne and Mercy Hospital for Women, Heidelberg, Australia [6]Department of Obstetrics and Gynaecology, The Ritchie Centre, School of Clinical Sciences, Monash University and The Hudson Institute of Medical Research, Clayton, Australia

Correspondence: natasha.dealwis@unimelb.edu.au

long-term maternal follow-up would provide an opportunity to test potential interventions in pregnancy and after birth, to determine whether it is possible to reduce the increased cardiovascular disease risk these individuals face—particularly important because very few interventions to reduce long-term risk have been studied (24).

In this article, we aimed to assess whether a popular animal model of preeclampsia could recapitulate the long-term effects of preeclampsia on maternal cardiovascular health. The model described in this article is an adaptation of the L-N[G]-Nitro arginine methyl ester (L-NAME) model of preeclampsia. L-NAME works by inhibiting nitric oxide synthase, the enzyme responsible for producing nitric oxide, a molecule involved in many biological processes including endothelium-dependent vasodilation (25). Reduction of nitric oxide bioavailability may contribute to vascular dysfunction in preeclampsia (26, 27). Consequently, nitric oxide donors are being considered potential candidates to examine for the treatment of preeclampsia (28, 29).

L-NAME was initially used to model preeclampsia in rats, inducing increased blood pressure, proteinuria, thrombocytopenia and fetal growth restriction (22). Since then, L-NAME has also been used to induce a preeclampsia-like phenotype in *mice*, inducing increased blood pressure, proteinuria, fetal growth restriction, impairment of placental morphology and other major organ and vascular alterations (30, 31, 32, 33, 34, 35, 36). However, these murine studies using L-NAME differ in their protocols, using varied concentrations of L-NAME, routes of administration, timing of exposure, and murine strains, that could respond differently to L-NAME (37). Furthermore, though L-NAME is commonly used in animal models to induce a preeclampsia-like phenotype, not many have followed up the animals post-delivery. Most of the studies that do, follow the offspring (38, 39, 40, 41, 42, 43). Only one study has examined maternal outcomes in the context of modelling long-term outcomes of a preeclamptic pregnancy, but it focused on cognitive deficit, not cardiovascular disease (44).

Therefore, in this study, we established the L-NAME model of preeclampsia in CBA x C57BL/6 (F1) mice. We assessed whether this model could mimic key pathways central to the pathogenesis of preeclampsia, as seen in other L-NAME models. Furthermore, we examined whether this model could mimic the increased maternal risk of cardiovascular disease following delivery, which is synonymous with preeclampsia.

## Results

### L-NAME administration to pregnant mice increased blood pressure in pregnancy

Hypertension is a key clinical characteristic of preeclampsia. Hence, we first assessed if L-NAME administered in our model could alter blood pressure. L-NAME administration significantly increased mean arterial blood pressure at E14.5 (control 103.7 ± 20.89, L-NAME 126.5 ± 20.92; mean ± SD) (*P* < 0.0001; Fig 1A) and E17.5 (control 111.9 ± 22.66, L-NAME 130.1 ± 18.52; mean ± SD) (*P* = 0.0001; Fig 1B). Both systolic and diastolic blood pressures were significantly increased

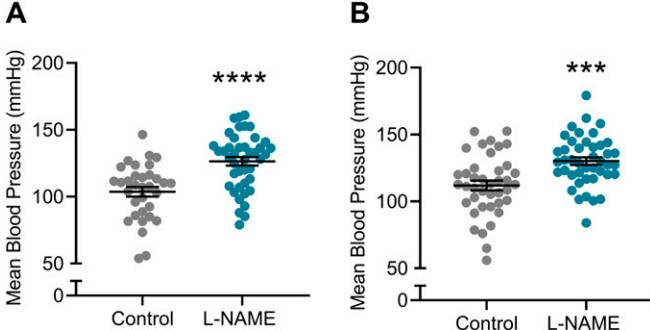

**Figure 1.  Effect of L-NAME administration on mean arterial blood pressure E14.5 and E17.5 of pregnancy.**
Blood pressure was measured by tail cuff plethysmography, (A) E14.5 mean blood pressure, (B) E17.5 mean blood pressure. **(A, B)** L-NAME administration significantly increased mean blood pressure from control levels at both time points. Data presented as mean ± SEM. Control n = 33–40, L-NAME n = 42–43. An unpaired *t* test was used to assess statistical differences between L-NAME and control group at each time point; ***P* < 0.001 *****P* < 0.0001.

at E14.5 (*P* < 0.0001; Fig S1A and B) and E17.5 (*P* = 0.0002 and *P* = 0.0001, respectively, Fig S1C and D).

### L-NAME administration in pregnancy did not induce proteinuria but may impact kidney structure at microscopic level

Proteinuria is evidence of kidney damage in preeclampsia. L-NAME administration did not significantly alter albumin to creatinine ratio at either E14.5 or E17.5 (Fig 2A and B). Furthermore, L-NAME did not alter the expression of genes associated with renal function, or known to be dysregulated with dysfunction (*Ccr2*, *Ctgf*, *Fn1*, *Hsd11b2*, *Il-1β*, *Mmp2* (expressed as ratio to expression of inhibitor, *Timp1*), *Mmp9* (expressed as ratio to expression of inhibitor *Timp1*), *Nhe1*, *Nlrp3*, *Nox2*, *Nox4*, *Nr3c1*, *Nr3c2*, *Sccn1a*, *Sgk1*, *Tgfβ1*, *Tgfβ2*, *Tgfβ3*, *Tnf*, and *Vcam1*) (Fig S2).

However, histological analysis revealed features in L-NAME mice that could be pathological in nature. In regions, there appeared to be haemoglobin and hyaline casts with inflammatory infiltrate and structural changes to the surrounding tubules (45). There also appeared to be increased cellularity and swelling of several glomeruli, and narrowing of the Bowman's space, compared with the control mice. This could be a sign of the renal pathology seen in individuals whose pregnancies are complicated by preeclampsia (46, 47). Haemoglobin-type casts were also visible in the control mouse kidneys; however, these are not surrounded by inflammatory or irregular cellular patterns and are likely to be artifacts (Fig 2C–F).

### L-NAME administration impairs fetal and placental growth

Placental dysfunction in preeclampsia can be associated with fetal growth restriction. Following L-NAME administration, we recorded a significant reduction in placental weight (*P* = 0.0423; Fig 3B) and fetal crown-to-rump length (*P* = 0.017; Fig 3D) corresponding to the dams culled at E17.5. However, neither fetal weight (Fig 3A), nor the fetal-to-placental weight ratio (Fig 3C) were altered. Pups of mice administered L-NAME had significantly reduced birthweight (*P* =

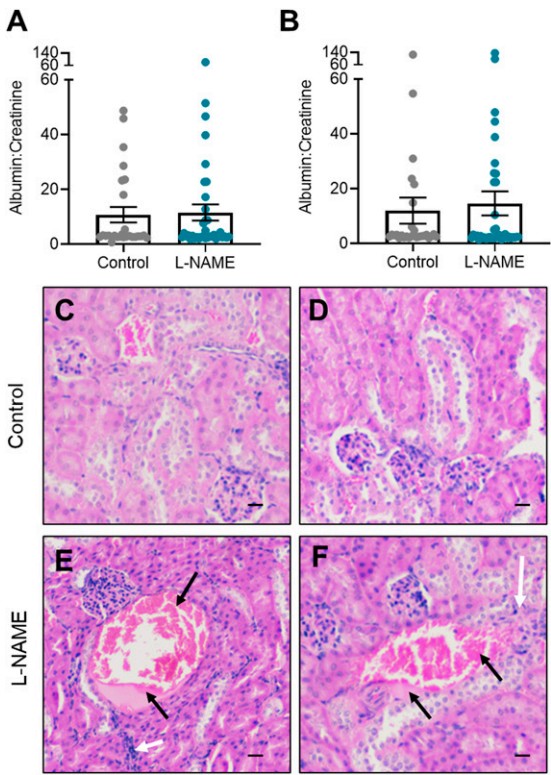

**Figure 2. Effect of L-NAME on urine albumin to creatinine ratio, and kidney structure in pregnancy.**

Maternal urine albumin and creatinine concentrations were measured via ELISA and enzymatic assay, respectively, at (A) E14.5 and (B) E17.5 of pregnancy. **(A, B)** Albumin to creatinine ratio was not significantly altered between the L-NAME and control mice. **(C, D)** Histology images of the cortex of PBS treated control mice kidneys show relatively normal glomeruli and tubules, with uniform staining. **(E, F)** Pathological features of L-NAME treated mice show inflammatory cell infiltration around regions of necrosis (black arrows) with haemoglobin (dark pink in section) and hyaline (light pink) casts that are surrounded by flattened nuclei and irregular cells (white arrows). Scale bar = 10 μm. Data presented as mean ± SEM. Control n = 26–28, L-NAME n = 36–39. A Mann–Whitney test was used to assess statistical differences in albumin:creatinine between L-NAME and control group at each time point.

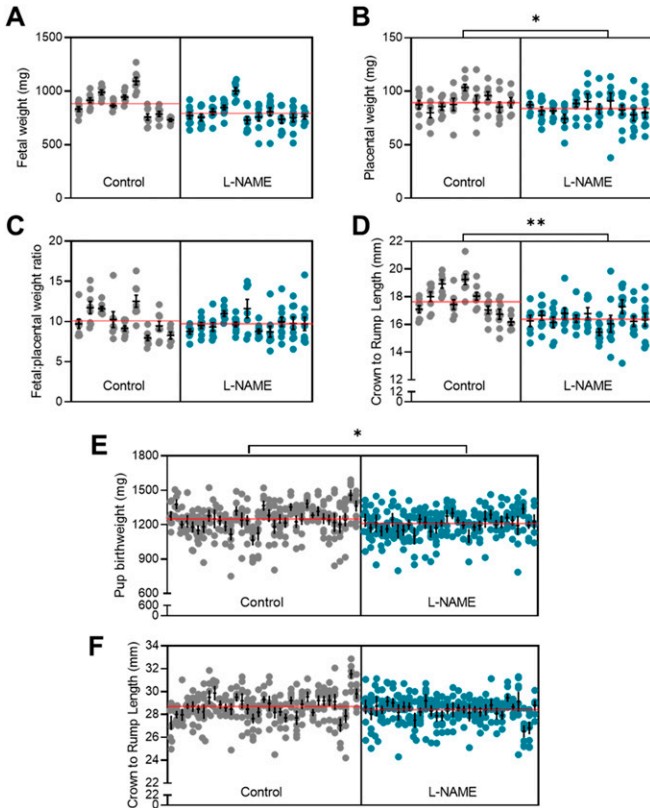

**Figure 3. Effect of L-NAME administration on E17.5 fetal, placental, and Day 1 pup size.**

(A) E17.5 fetal weight, (B) E17.5 placental weight, (C) E17.5 fetal to placental weight ratio, (D) E17.5 crown-to-rump length, (E) pup birthweight (F) pup crown-to-rump length at birth. **(A, B, C, D)** L-NAME administration in pregnancy significantly decreased E17.5 placental weight (B) and crown-to-rump length (D), but not fetal weight (A) or the fetal to placental weight ratio (C). **(E, F)** At birth, pups from mice administered L-NAME had reduced pup weight (E), with no significant change to crown-to-rump length (F). Data presented as mean ± SEM. E17.5 Litters: control n = 9, L-NAME n = 11. E17.5 fetuses: control n = 68, L-NAME n = 97. At birth, litters: Control n = 35, L-NAME n = 32. Pups: Control n = 274, L-NAME n = 246. Each column of points represents the fetuses, pups or placentas corresponding to each dam. Black line and error bars represent mean ± SEM within each litter. The red transparent line across each group represents the mean across all litters. Data were statistically analysed using a nested $t$ test; *$P < 0.05$, **$P < 0.01$.

0.0234; Fig 3E) with no significant difference in crown-to-rump length (Fig 3F) compared with control at birth. There was no significant difference in the overall litter size (data not shown). L-NAME administration did not alter gestational length—all mice gave birth by the morning of E19.5.

At E17.5, the placentas of mice administered L-NAME demonstrated significantly increased expression of the anti-angiogenic *Flt1* transcript ($P = 0.0049$; Fig 4B), and antioxidant *Hmox-1* ($P = 0.0193$; Fig 4G). L-NAME administration did not alter placental expression of *Vegfa*, *Plgf*, *Nos3*, *Tnf*, or *Vcam1* (Fig 4A and C–F). Placental structure was also studied for a subset of placentas. The placentas collected from mice given L-NAME did not have significantly altered cross-sectional area, or changes in the blood space, junctional zone, and labyrinth zone areas (Fig S3). Mean arterial blood pressure corresponding to the subset of mice used for these analyses (culled at E17.5) is presented in Fig S4.

## L-NAME administration in vivo increases circulating factors associated with preeclampsia in pregnancy, but does not alter ex vivo vascular reactivity

Next, we assessed levels of soluble factors known to be elevated in the maternal circulation with preeclampsia. L-NAME administration caused a significant increase in serum levels of the inflammatory marker, CRP ($P = 0.0205$; Fig 5A), anti-angiogenic factor, sFLT-1 ($P = 0.0238$; Fig 5B), and potent vasoconstrictor, ET-1 ($P = 0.0486$; Fig 5C).

Preeclampsia features vascular dysfunction, leading to systemic maternal hypertension. Administration of L-NAME throughout pregnancy did not alter vasoconstriction to phenylephrine (Fig 5D) or vasodilation to acetylcholine (Fig 5E), assessed in ex vivo wire myography experiments. There were no changes to LogEC50 (half-maximal response corresponding to the shift of the curve), area

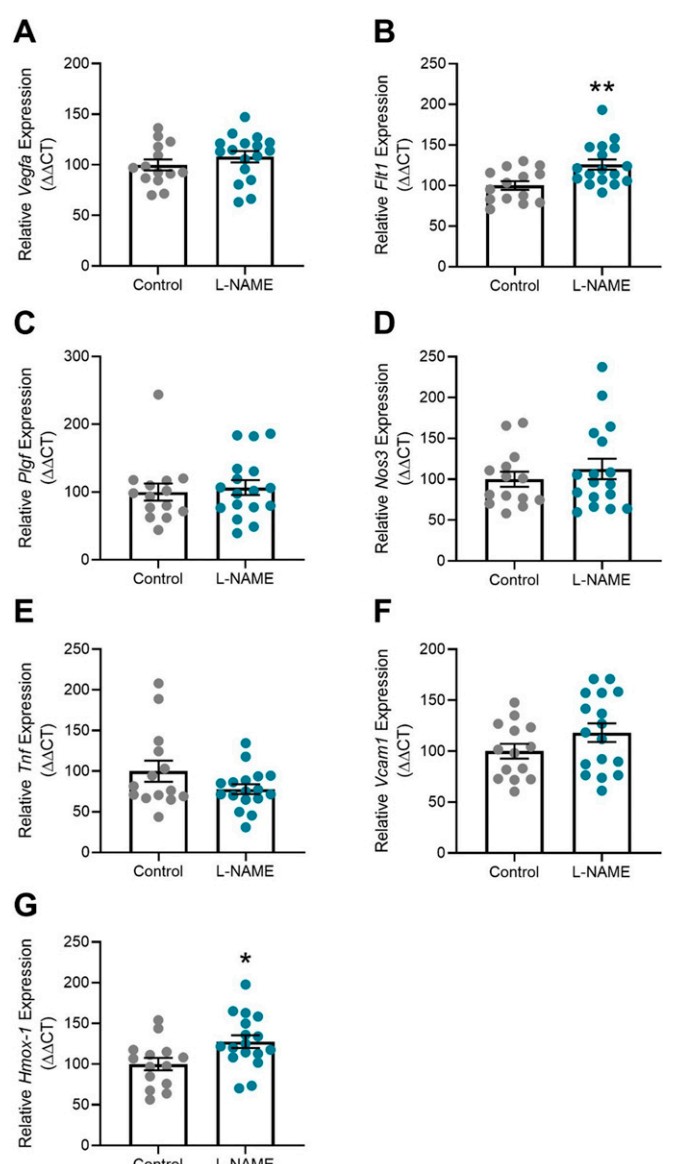

Figure 4.   Effect of L-NAME administration on placental gene expression at E17.5.
Gene expression was assessed via qPCR. **(B, G)** Expression of *Flt1* (B) and *Hmox-1* (G) were significantly increased in placentas of mice administered L-NAME compared with control. **(A, C, D, E, F)** L-NAME did not alter placental expression of *Vegfa*, *Plgf*, *Nos3*, *Tnf*, or *Vcam1*. Data are presented as mean ± SEM. Control n = 6 mice, L-NAME n = 10 mice. 1–3 placentas were chosen at random from each. Samples with low RNA yield at extraction were excluded. Data were statistically analysed using an unpaired *t* test (if in normal distribution) or Mann–Whitney test; *P < 0.05, **P < 0.01.

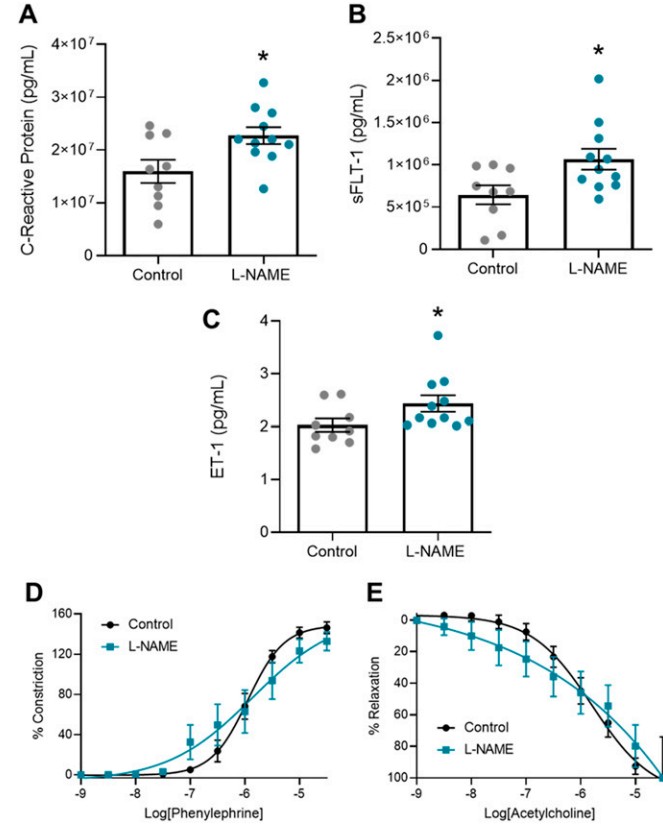

Figure 5.   Effect of L-NAME administration on circulating factors associated with preeclampsia and vascular reactivity at E17.5 of pregnancy.
Circulating factors were measured by ELISA. Vascular reactivity of mesenteric arteries was assessed by wire myography. **(D, E)** Vasoconstriction to phenylephrine (E) Vasodilation to acetylcholine. **(A, B, C)** C-reactive protein, (B) sFLT-1, and (C) ET-1 were significantly increased in the mice administered L-NAME compared with controls. There was no change in vascular reactivity with L-NAME administration. LogEC50, area under the curve and maximal response for these curves are presented in Fig S5. Data are presented as mean ± SEM. Control n = 7–9 mice, L-NAME n = 7–11 mice. Levels of circulating factors were statistically analysed using an unpaired *t* test (for normal distribution) or Mann–Whitney test; *P < 0.05. Vascular response at each dose of agonist was statistically analysed using mixed-effects analysis with Šidák correction for multiple comparisons.

under the curve (total response) or maximum response (Fig S5). The arteries from each group had no significant difference in response to the high potassium solution (100 mM KPSS; Fig S6).

Consistent with this, L-NAME administration did not alter mesenteric artery expression of the enzyme responsible for production of nitric oxide *Nos3*, the endothelial dysfunction marker *Vcam-1*, or the receptors for vasoconstrictor endothelin-1, *Ednra* and *Ednrb* (Fig S7).

## L-NAME administration in pregnancy does not alter maternal blood pressure, or circulating levels of factors associated with endothelial dysfunction up to 10 wk post-delivery

We assessed whether the preeclampsia-like phenotype induced by L-NAME could alter long-term blood pressure, as individuals have a higher risk of hypertension after a pregnancy complicated by preeclampsia (48). Despite mice administered L-NAME experiencing elevated blood pressure during pregnancy, their diastolic, systolic, and mean arterial blood pressure returned to control levels following discontinuation of L-NAME treatment at 1 wk post-delivery, and remained at control levels up to 10 wk post-delivery (Figs 6A and S8). Furthermore, circulating levels of ET-1 and CRP also returned to control levels at 1 wk post-delivery, and remained so up to 10 wk post-delivery (Fig 6B and C).

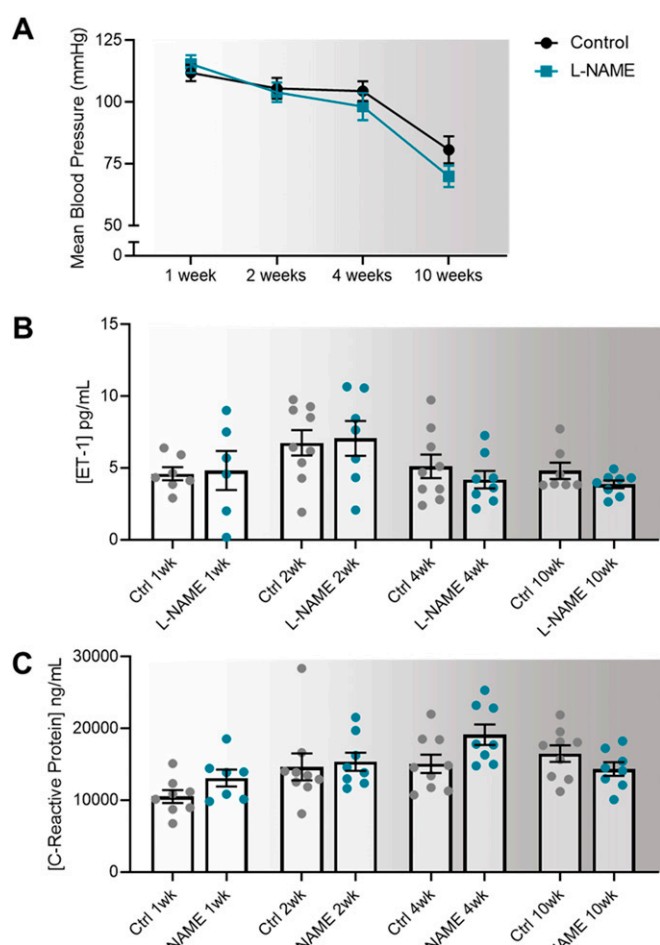

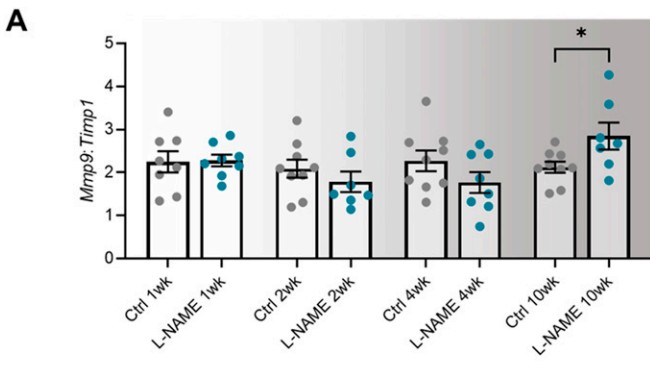

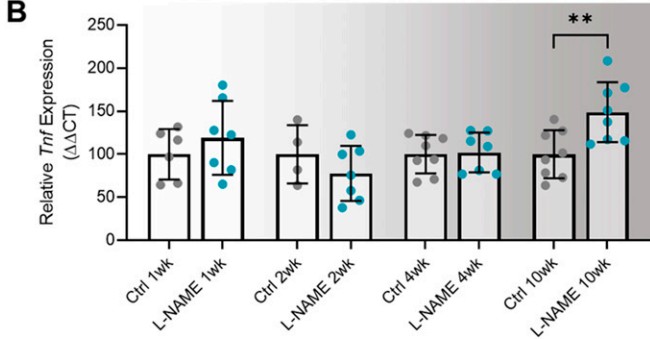

**Figure 6. Effect of L-NAME administration in pregnancy on mean arterial blood pressure, and serum levels of endothelin-1 and C-reactive protein at 1-,2-, 4-, and 10-wk post-delivery.**

Blood pressure was measured via tail cuff plethysmography and circulating factors by ELISA. **(A, B, C)** L-NAME administration in pregnancy did not alter post-delivery (A) blood pressures, or circulating (B) ET-1 and (C) C-reactive protein levels at any time point. Data presented as mean ± SEM. Blood pressures–Control: 1 wk n = 35, 2 wk n = 27, 4 wk n = 15, 10 wk n = 9; L-NAME: 1 wk n = 32, 2 wk n = 24, 4 wk n = 13, 10 wk n = 8. ELISAs were carried out using serum samples from n = 6–9 mice/group at each time point. Data were statistically analysed between the L-NAME and control group at each time point with a unpaired *t* test (if in normal distribution) or Mann–Whitney test.

### L-NAME administration in pregnancy does not alter post-delivery urine albumin to creatinine ratio, but is associated with up-regulation of *Mmp9:Timp1* at 10 wk post-delivery

Consistent with findings during pregnancy, mice administered L-NAME did not have proteinuria (altered urine albumin to creatinine ratio) at any time point post-delivery (Fig S9). However, we found that *Mmp9:Timp1* expression was elevated in the mice given L-NAME, at 10 wk post-delivery ($P$ = 0.0418; Fig 7A). *Mmp2* expression was also elevated at 10 wk post-delivery ($P$ = 0.0229; Fig S10F), along with *Mmp9* expression alone ($P$ = 0.0311; Fig S10H); however, the ratio of *Mmp2* to its inhibitor *Timp1* was not altered (Fig S10G). There were no other changes in expression of genes associated with renal function and dysfunction (*Ccr2, Ctgf, Fn1, Hsd11b2, IL-6, Nhe1, Nlrp3,*

**Figure 7. Effect of L-NAME administration in pregnancy on long-term expression of markers of dysfunction in the maternal heart and kidney.**
**(A)** Kidney expression of *Mmp9* presented as a ratio to *Timp1* (its inhibitor) expression. **(B)** Heart expression of *Tnf*. Relative expression at 1-,2-,4-, and 10-wk post-delivery was assessed by qPCR. Kidney *Mmp9:Timp1* and heart *Tnf* expression were not altered in mice administered L-NAME in pregnancy compared with control at 1-, 2-, 4-, or 10- wk post-delivery. At 10 wk post-delivery, mice administered L-NAME in pregnancy had significantly increased kidney *Mmp9:Timp1* expression and increased heart *Tnf* expression. Data are presented as mean ± SEM. Data were statistically analysed between the L-NAME and control group at each time point with a unpaired *t* test (if in normal distribution) or Mann–Whitney test; *$P$ < 0.05, **$P$ < 0.01. n = 7–9 kidneys/group, n = 4–8 hearts/group at each time point.

*Nox2, Nox4, Nr3c1, Nr3c2, Scnn1a, Sgk1, Tgfβ1, Tgfβ2, Tgfβ3, Timp1, Tnf,* or *Vcam-1*; Fig S10A–E and I–V).

### L-NAME administration in pregnancy is associated with increased *Tnf* expression in maternal hearts at 10 wk post-delivery

We examined gene expression in hearts of the dams post-delivery to assess potential long-term effects on cardiac function. We found that *Tnf* expression was significantly increased in the hearts of L-NAME treated mice at 10 wk post-delivery ($P$ = 0.0078), but not at 1-, 2-, or 4-wk post-delivery (Fig 7B). There was no change in expression of the other cardiac genes assessed (*Bnp, Camk2a, Ccr2, Ctgf, Il-6, Mmp2, Mmp9 Nlrp3, Nox2, Nr3c1, Nr3c2, Tgfβ1, Tgfβ2, Tgfβ3, Timp1 Vcam-1, Mmp2:Timp1,* and *Mmp9:Timp1*; Fig S11).

### L-NAME administration in pregnancy is associated with increased mesenteric artery vasoconstriction to phenylephrine at 10 wk post-delivery

To assess the long-term effects to the vasculature, we also examined vascular reactivity of mesenteric arteries 1-, 2-, 4-, and 10-wk post-

delivery. L-NAME administration in pregnancy had no effect on mesenteric artery vasoconstriction to phenylephrine at 1-, 2-, and 4-wk post-delivery (Fig 8A–C). However, at 10 wk post-delivery, the arteries from the mice treated with L-NAME in pregnancy constricted significantly more to $10^{-6}$ M phenylephrine than control ($P = 0.0178$; Fig 8D). L-NAME administration in pregnancy did not alter vasorelaxation to acetylcholine at any time point post-delivery (Fig 8E–H). There were no changes in LogEC50, AUC or maximal response ($E_{max}$–maximum constriction; $R_{max}$–maximum relaxation) at any time point to either phenylephrine or acetylcholine (Figs S12 and S13). The arteries from each group had no significant difference in response to the high potassium solution at each time point (100 mM KPSS; Fig S14).

We also assessed expression of genes altered in vascular dysfunction. There were no changes in vascular expression of *Nos3*, *Vcam1*, *Ednra*, and *Ednrb* at any time point post-delivery (Fig S15).

## Discussion

In this study, we established the L-NAME mouse model of preeclampsia in (CBA x C57BL/6) F1 mice, and assessed whether this model could recapitulate the long-term effects of preeclampsia on maternal cardiovascular health. During pregnancy, L-NAME administration markedly increased blood pressure, increased circulating ET-1, CRP, and sFLT-1 levels, and restricted fetal growth. Despite recapitulating key aspects of preeclampsia in the mouse, there were no major long-term cardiovascular impacts postpartum. Except, at 10 wk post-delivery, we observed increased expression of *Tnf* and *Mmp9:Timp1* in the maternal heart and kidney, respectively, and moderately enhanced vasoconstriction.

L-NAME, a potent vasoconstrictor, enables the simulation of the systemic vasoconstriction that occurs in preeclampsia. This vasoconstriction is likely responsible for the impaired fetal and placental growth detected in our model. Ma et al have previously shown that L-NAME administration initiated in early gestation is able to induce feto-placental unit impairment, compared with administration started late in pregnancy (34). We suggest that early administration of L-NAME in our model leads to perturbations during early placental development, given we continue to observe growth restriction even after cessation of L-NAME. Moreover, we suggest the vasoconstriction caused by L-NAME, reduced blood flow to the feto-placental unit, restricting fetal growth. These early changes are especially relevant to early-onset preeclampsia, which is associated with higher rates of adverse fetal outcomes and placental insufficiency (49). Although pup and placental weight were both reduced, fetal:placental weight ratio was not altered, and placental cross-sectional area was also unchanged. This suggests that any impairments in placental capacity to pass nutrients to the fetus were not due to structural changes of the placenta. Rather, the fetal growth impairment may be due to reduced total blood supply due to vasoconstriction of the uterine arteries. Further studies are required to uncover the mechanisms behind this growth impairment, including whether there are any clinical parameters of interest including fetal brain sparing.

Interestingly, L-NAME increased placental expression of *Flt1* and circulating sFLT-1. sFLT-1 is known to be increased in the circulation of individuals with pregnancies complicated by preeclampsia (50). This clearly demonstrates that administration of L-NAME caused an up-regulation of this key pathway, offering a compelling model of preeclampsia. In other studies, L-NAME rodent models demonstrate mixed results for changes in Flt1 and its soluble circulating form, although some of these discrepancies may be due to the varied L-NAME concentrations and routes of administration used (31, 51, 52, 53, 54, 55, 56, 57, 58, 59, 60, 61, 62). Complex viral vector

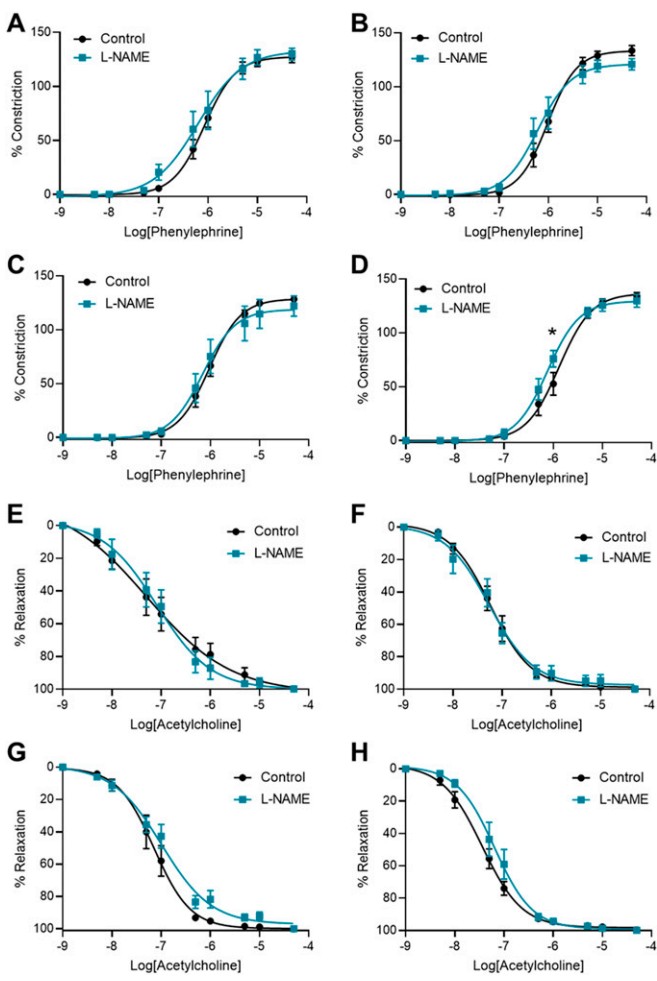

**Figure 8. Effect of L-NAME administration in pregnancy on vascular reactivity post-delivery.**
Effect of L-NAME administration in pregnancy on vascular reactivity measured in mesenteric arteries collected at 1 wk (A, E), 2 wk (B, F), 4 wk (C, G), and 10 wk (D, H) post-delivery. Vascular reactivity was assessed using ex vivo wire myography, assessing vasoconstriction to phenylephrine and vasodilation to acetylcholine. **(A, B, C)** Arteries collected from the mice administered L-NAME in pregnancy did not respond differently to phenylephrine compared with controls at the 1-, 2-, and 4-wk time points post-delivery. **(D)** At 10 wk post-delivery, the arteries collected from the mice that were administered L-NAME in pregnancy constricted significantly more than the control vessels at $10^{-6}$ M phenylephrine. **(E, F, G, H)** The vessels collected from the mice administered L-NAME in pregnancy did not have altered vasodilation to acetylcholine at any time point post-delivery. LogEC50, area under the curve and maximal response for these curves are presented in Figs S12 and S13. Data presented as mean ± SEM. n = 7–9 mice/group at each time point. Vascular response at each dose of agonist were statistically analysed using mixed-effects analysis with Šidák correction for multiple comparisons.*$P < 0.05$.

models inducing placental specific *sFLT-1* overexpression have been developed to model the increased placental secretion of sFLT-1 akin to that observed in preeclampsia in women (19, 63). However, a strength of the model we demonstrate here is this model provides a cheaper, simpler and more accessible way to induce key characteristics of preeclampsia in a small animal model, with ability to study promising candidate drugs to mitigate key aspects of preeclampsia pathogenesis.

Here, we also found that mice receiving L-NAME demonstrated increased placental expression of *Hmox-1*, a cytoprotective anti-oxidant enzyme. We hypothesise that this may have been triggered by tissue hypoxia as a result of markedly reduced blood flow, as has been observed in other rat, mouse, bovine and monkey models of hypoxia (64, 65), perhaps an adaptation to mediate oxidative stress in the placenta (contrasting to in human cells, where hypoxia is a repressor of Hmox-1 (64)). However, this contrasts with other studies using L-NAME in rats, where they found placental Hmox-1 protein was reduced with L-NAME administration (54, 66, 67, 68). Studies examining the effects of L-NAME on placental oxidative stress in *rats* have shown mixed results (68, 69, 70, 71, 72). In L-NAME *mouse* models, placental Hmox-1 levels have not previously been evaluated, though one study reports that L-NAME is associated with fatty acid oxidation (32). Examination of the duration of L-NAME administration and evidence of oxidative stress levels would help determine the effect of L-NAME on placental oxidative stress and Hmox-1. This is of particular interest to our group, as we have previously demonstrated that HMOX-1 is not altered in human preeclamptic placental tissue (73).

L-NAME treatment did not induce evident signs of proteinuria in the pregnant dams, as measured by albumin to creatinine ratio (although we note this is not a clinical grade test). Increased albumin concentration in the urine is associated with kidney damage and renal injury, with normalisation to creatinine (consistently released). However, histological assessment of the kidneys demonstrated signs of pathology in the mice given L-NAME during pregnancy. It is possible that the kidney damage was not severe enough to cause proteinuria in this model. Others have reported L-NAME in mouse models demonstrate proteinuria, but these studies have based this finding on altered total urine protein (32, 35, 36), albumin alone (33), or creatinine alone (31), and not the albumin to creatinine ratio, which is a better measure, and used clinically. Furthermore, we did not find changes in expression of genes associated with kidney damage or dysfunction.

Elevated circulating levels of the potent vasoconstrictor ET-1, inflammatory marker CRP and anti-angiogenic factor sFLT-1 in our L-NAME model mirror the increase found in the circulation of individuals with preeclampsia (50, 74, 75). Intriguingly, this was not accompanied with a change in mesenteric artery vascular reactivity. We propose that this may be because L-NAME may not have a permanent effect on the vasculature, but rather works through its ability to cause vasoconstriction through decreasing production of nitric oxide. Hence, when the arteries are no longer exposed to L-NAME in an ex vivo system outside the body, nitric oxide levels can recover. Although L-NAME is commonly directly applied to vessels in ex vivo studies (used for over 20 yr (76)), few studies have explored the effect of administering L-NAME in vivo on vascular reactivity. One study showed that L-NAME administration impaired

acetylcholine induced vasorelaxation in rat aortic rings (77) and another that L-NAME impaired myogenic tone of rat uterine radial arteries (78). However, it is understood that vascular adaptations in pregnancy are distinct between rats and mice (79), which may explain the discrepancy.

In this study, we chose to assess vascular reactivity in mesenteric arteries, as they are resistance arteries involved in control of blood pressure (80, 81), and allowed us to assess *systemic* changes in pregnancy (that can be compared with post-pregnancy responses) (79). However, future studies may benefit from investigation of uterine artery vascular reactivity to help understand the mechanisms behind the impaired placental and pup size. In addition, phenylephrine and acetylcholine were used to assess vascular reactivity as their receptors are present on mesenteric arteries, and they are commonly used to assess reactivity in the myograph field. However, it may be beneficial to assess the response to other agonists, including the vasoconstrictor, ET-1, which we know is elevated in the circulation in this model, and in individuals with preeclampsia (82).

An important part of this study was also to investigate whether the L-NAME-induced changes could impact long-term maternal cardiovascular health—an important consideration currently lacking in many established models of preeclampsia. Preeclampsia and gestational hypertension are associated with increased postpartum maternal cardiovascular disease risk (10, 11, 12, 13, 14). Hence, we expected that mice administered L-NAME would have altered cardiovascular disease risk indices post-pregnancy.

Because blood pressure returned to control levels within 1 wk post-delivery, we suspect that there could be recovery in nitric oxide levels with the cessation of L-NAME administration, permitting vasodilation and reducing blood pressure. After being elevated in pregnancy, the potent vasoconstrictor ET-1 and the inflammatory mediator CRP both returned to control levels post-delivery with L-NAME cessation. As these factors play a role in enhancing endothelial dysfunction, it is likely that their return to control levels also contribute to recovery of blood pressure. This is unlike what happens after a hypertensive pregnancy, where blood pressure often remains elevated, persisting more than 20 yr post-pregnancy (48), and where endothelial dysfunction persists (83, 84).

Proteinuria in preeclampsia can persist post-delivery, although some studies suggest this is resolved by 2 yr post-delivery (85, 86, 87). As L-NAME did not induce proteinuria in our animals during pregnancy, we did not expect long-term changes in the kidney post-delivery. However, at 10 wk post-delivery, the mice administered L-NAME in pregnancy had increased renal *Mmp9:Timp1* expression. Mmp9 is a matrix metalloproteinase that (in balance with its inhibitor, Timp1) is involved in cleavage of extracellular matrix and other proteins, important for normal kidney function and structure (88). Studies have also shown Mmp9 has a role in inflammation (89, 90, 91). The elevation in *Mmp9:Timp1* expression may imply an increase in inflammation in the kidneys, but further molecular studies are needed to investigate this.

Studies have observed increased cardiac dysfunction (left ventricular hypertrophy) at 1 yr post-preeclampsia (12), and an association between preeclampsia and increased risk of admission to hospital or death due to ischemic heart disease (92). Interestingly, at 10 wk post-delivery (equivalent to ~9.5 yr in human) *Tnf*

mRNA expression was elevated in the hearts of mice administered L-NAME in pregnancy, implying increased cardiac inflammation over time. Further studies to determine if there is an increase in Tnf protein production and other markers of inflammation are needed to confirm this. Exploration of the cardiac structure and function may be helpful in evaluating whether our model produces the same left ventricular dysfunction seen post-preeclampsia.

Because persistent endothelial dysfunction is likely to contribute to increased risk of cardiovascular disease, we assessed whether vascular reactivity may be altered post-delivery. Although there were no changes in vascular reactivity at E17.5 gestation, we found increased mesenteric artery constriction at 10 wk post-delivery at one dose of phenylephrine. This would imply that L-NAME administration may have exerted some permanent effects, either directly or indirectly (as the hypertensive phenotype may have driven other indirect changes that required time to become apparent), and further studies would be required to examine this. Increased sensitivity to vasoconstrictors would be considered detrimental; however, we do not know how functionally relevant the moderate difference we detected may be. It could be a sign of increased arterial wall stiffness as demonstrated in a study examining mice at 4 wk post-partum (93). Assessing vascular responses to other agonists may provide additional insight into other pathways that may be altered by L-NAME.

We did not observe significant changes in expression of inflammatory, or endothelial dysfunction-related genes in the mesenteric arteries post-delivery. However, there are limited data as the samples had to be pooled, leading to a low effective sample size. Increasing the sample size and examining protein levels would be valuable to discern the expression of endothelial dysfunction-relevant genes post-delivery.

The long-term changes identified here are far less adverse than what we expected, based on the increased cardiovascular risk known post-preeclampsia. In our study, we chose time points up to 10 wk postpartum, modelling the human equivalent of ~1, 2, 4, and 9.5 yr post-delivery. Expanding the study to include later time points (beyond 10 wk post-delivery) may have uncovered further cardiovascular changes, as another model has shown (94). However, the time points used in this study were chosen to reflect the clinical phenomena, examining earlier time points akin to human clinical observations, as individuals demonstrate impaired cardiovascular function soon after a pregnancy complicated by preeclampsia (13, 48).

As the L-NAME model is thought to mainly cause reduced nitric oxide levels driving systemic vasoconstriction, the preeclampsia phenotype observed here may not be as severe as that in other models, including knockout models (95). Indeed, our study highlights that L-NAME administration started in pregnancy alone is unable to induce the severe long-term phenotype expected. In the preeclampsia field, a dogma exists to whether those who get preeclampsia already have predisposition for cardiovascular disease (96), or whether pregnancy acts as the "stress test," exacerbating this risk (97). Here we selected mice that should not have an underlying predisposition for cardiovascular disease, nor stressors that could exacerbate risk of cardiovascular disease post-delivery. Assessment using a strain that is more sensitive to L-NAME may demonstrate a more persistent chronic effect on the cardiovascular

system (37). In addition, it would be interesting to use a strain or genetic model with increased cardiovascular disease risk, potentially an obese or high salt diet mouse model, or a strain genetically modified to enhance vascular dysfunction, mimicking predisposition to preeclampsia and long-term cardiovascular disease. Some studies have also used a cardiovascular injury event post-delivery to assess whether the preeclamptic pregnancy may make the system more sensitive to injury (98). Another important aspect would be to assess whether lactation may have an effect on cardiovascular health post-hypertensive pregnancy, as studies have shown that lactation may reduce vasocontractility, enhance vasorelaxation and increase vessel distensibility (99). Here, the dams were removed from the pups following birth, and thus this would constitute a higher likelihood of hypertension and vascular dysfunction if this association held in this model.

## Conclusion

Preeclampsia is a complex, multi-organ condition and is still not completely understood. We acknowledge that it is unlikely that one model alone would be able to cover all aspects of preeclampsia pathogenesis. However, the L-NAME model described here provides a relatively simple small animal model of preeclampsia that can be used to assess the potential beneficial effects of candidate therapeutics that can target and mitigate systemic vascular constriction. However, it does not amply model the long-term cardiovascular effects seen in individuals soon after a pregnancy complicated by preeclampsia. Knowing that preeclampsia is associated with increased risk of cardiovascular disease post-pregnancy, it is essential that we develop precise models of its long-term effects and develop interventions that not only improve obstetric outcomes, but also improve long-term health.

# Materials and Methods

### Animal studies

Animal experiments were approved by the Austin Health Animal Ethics Committee (A2018/05596) and followed the National Health and Medical Research Council ethical guidelines for the care and use of animals for scientific purposes. To have sufficient power for the final time points, a larger cohort of mice were required. 3-wk-old CBA x C57BL/6 (F1) female mice (n = 87) were sourced from the Florey Institute of Neuroscience and Mental Health (The University of Melbourne). Mice were group-housed in conventional open-top cages, on a 12-h light/dark cycle, with food and water available ad libitum (18–22°C; 50% relative humidity). Before experimentation, mice were acclimated to the CODA non-invasive blood pressure system (Kent Scientific) in a seven-stage process involving exposure to the restraining tube and tail cuff.

From 6 to 16 wk of age (equivalent to ~15–25 yr in human equivalent (100)), F1 females were mated overnight with stud F1 male mice. Pregnancy was confirmed by the presence of a copulatory plug the following morning; designated as embryonic day (E)0.5.

### L-NAME mouse model of preeclampsia

L-NAME (50 mg/kg/day; Sigma-Aldrich (30, 101)) or phosphate buffered saline (PBS; 137 mM NaCl, 10 mM $Na_2HPO_4$, 1.8 mM $KH_2PO_4$, and 2.7 mM KCl, pH 7.4) as control was administered daily from embryonic day (E)7.5 to E17.5 of pregnancy (approximately early second trimester to term in human equivalent to model early onset disease) via 100 µl subcutaneous injection. On E14.5 and E17.5 of pregnancy, urine was collected (spot collection on/upon handling), and blood pressure measured.

After blood pressure measurement on E17.5 a subset of mice (control n = 9, L-NAME n = 11) were anesthetized with 5% isoflurane in oxygen, and cardiac puncture performed to collect maternal blood. Mice were then culled by cervical dislocation. Blood samples were allowed to coagulate at room temperature before centrifugation to separate the serum fraction, which was snap frozen and stored at –80°C. Maternal kidneys were either fixed overnight in 10% neutral buffer formalin or preserved in RNAlater. Fetuses were counted and weighed, and crown-to-rump length measured (with digital calipers). Placentas were also collected, weighed, and preserved in RNAlater (Invitrogen), or fixed in 10% neutral buffered formalin. Tissues remained in RNAlater for a minimum of 48 h (as per the manufacturer's recommendations), then snap frozen and stored at –80°C until subsequent analysis. Formalin-fixed tissue was embedded in paraffin and sectioned on a microtome for examination of structural changes. The intestinal tract was collected in ice cold PBS for dissection of mesenteric arteries for vascular studies.

All remaining mice (control n = 35, L-NAME n = 32) were allowed to litter naturally for long-term follow-up of dams post-delivery. Day (D)1 pups were counted, weighed, and crown-to-rump length recorded before euthanasia by decapitation. Long-term maternal urine collection, blood pressure measurements and culls were performed at 1-, 2-, 4-, and 10-wk post-delivery. At cull (anaesthesia and cervical dislocation), cardiac puncture was performed and urine, maternal organs (heart and kidney collected in RNAlater) and intestinal tract collected as described above.

### Vascular reactivity studies

Second order mesenteric arteries were carefully dissected from surrounding connective and adipose tissue in Krebs physiological salt solution (NaCl 120 mM, KCl 5 mM, $MgSO_4$ 1.2 mM, $KH_2PO_4$ 1.2 mM, $NaHCO_3$ 25 mM, D-glucose 11.1 mM, and $CaCl_2$ 2.5 mM). Dissected arteries (2 mm length) were then mounted on the 620M Wire Myograph (Danish Myo Technology [DMT]) using 25-µm-diameter gold-plated tungsten wires (W005230; Goodfellow), and bathed in Krebs solution continuously bubbled with carbogen (95% $O_2$, 5% $CO_2$), and warmed to 37°C. The arteries were normalised to 100 mm Hg (13.3 kPa) pressure using the DMT normalisation module on LabChart software (ADInstruments) with IC1/1C100 = 1. Smooth muscle viability was confirmed using high potassium physiological salt solution (KPSS; NaCl 25 mM, KCl 100 mM, $MgSO_4$ 1.2 mM, $KH_2PO_4$ 1.0 mM, $NaHCO_3$ 25 mM, D-glucose 11.1 mM, and $CaCl_2$ 2.5 mM). Endothelial function was assessed by pre-constricting arteries to 50–70% of maximal constriction to KPSS with phenylephrine (Sigma-Aldrich), then relaxed with the endothelial dependent dilator, acetylcholine

(Sigma-Aldrich). Confirmation of response to KPSS, and greater than 80% relaxation was required for inclusion of the vessel. Constriction and relaxation dose response curves were then generated using phenylephrine and acetylcholine ($10^{-9}$ to $10^{-4.5}$M).

After collection of second order mesenteric arteries for vascular studies, the remaining mesenteric arteries were excised and preserved in RNAlater as described above.

### Quantitative polymerase chain reaction (qPCR)

RNA was extracted from the RNAlater preserved placenta, heart (whole), kidney, and mesenteric arteries using the GenElute Mammalian Total RNA Miniprep Kit (Sigma-Aldrich) and quantified with a NanoDrop 2000 spectrophotometer (Thermo Fisher Scientific). Extracted RNA was converted to cDNA using the Applied BiosystemsTM High-Capacity cDNA Reverse Transcription Kit, as per manufacturer guidelines on the iCycler iQ5 (Bio-Rad).

Quantitative PCR with Taqman reagents was performed to quantify mRNA expression using primers purchased from Life Technologies. Primers are listed in Table 1. Quantitative PCR was performed on the CFX384 (Bio-Rad) with the following run conditions: 50°C for 2 min; 95°C for 20 s, 95°C for 3 s, 60°C for 30 s (40 cycles). All expression data were normalised to expression of reference gene (chosen based on their stability in each tissue) as an internal control and calibrated against the average Ct of the control samples, with each biological sample run in technical duplicate.

### ELISA

Concentrations of soluble fms-like tyrosine kinase 1 (sFLT-1), endothelin-1 (ET-1), and C-reactive protein (CRP) in maternal serum were measured using the Mouse sVEGFR1/Flt-1 DuoSet ELISA kit (samples diluted 1:100), Mouse Endothelin-1 Quantikine ELISA Kit (sensitivity 0.207 pg/ml), and Mouse C-Reactive Protein/CRP Quantikine ELISA Kit (sensitivity 0.015 ng/ml) (R&D Systems), respectively, according to the manufacturer's instructions (inter and intra assay coefficients of variation under 10%).

### Urine albumin to creatinine assessment

Albumin and creatinine concentrations in urine were measured using the Mouse Albumin ELISA kit (sensitivity 5 ng/ml) and Mouse Creatinine Kit (80630, 80350; Crystal Chem) run according to manufacturer's instructions.

### Maternal kidney and placenta histology

After fixation in 10% neutral buffered formalin overnight, placentas and kidneys were washed in PBS and embedded in paraffin for sectioning at 5 µM thickness. The sections were deparaffinized in xylene and rehydrated through descending grades of ethanol before haematoxylin and eosin staining. Kidney histology was visualized and captured using a Nikon Eclipse Ci microscope and camera at 100 µm magnification (n = 3 sections/condition, whole kidneys from two dams were analysed per group).

Placental sections were imaged using the ScanScope system (Aperio Technologies). General histological appearance was

**Table 1.  Primers used for Taqman PCR.**

| Gene name | Gene symbol | Taqman primer assay ID |
|---|---|---|
| C–C motif chemokine receptor 2 | Ccr2 | Mm04207877_m1 |
| Calcium/calmodulin-dependent protein kinase II alpha | Cam2kα | Mm00437967_m1 |
| Connective tissue growth factor | Ctgf | Mm01192933_g1 |
| Endothelin receptor A | Ednra | Mm01243722_m1 |
| Endothelin receptor B | Ednrb | Mm00432989_m1 |
| Fibronectin 1 | Fn1 | Mm01256744_m1 |
| FMS-like tyrosine kinase 1 | Flt1 | Mm00438980_m1 |
| Hemoxygenase 1 | Hmox-1 | Mm00516005_m1 |
| Hydroxysteroid 11-beta dehydrogenase 2 | Hsd11b2 | Mm01251104_m1 |
| Interleukin-1 beta | Il-1β | Mm00434228_m1 |
| Interleukin-6 | Il-6 | Mm00446190_m1 |
| Matrix metallopeptidase 2 | Mmp2 | Mm00439498_m1 |
| Matrix metallopeptidase 9 | Mmp9 | Mm00442991_m1 |
| NADPH oxidase 2 | Nox2 | Mm01287743_m1 |
| NADPH oxidase 4 | Nox4 | Mm00479246_m1 |
| Natriuretic peptide type B | Bnp | Mm01255770_g1 |
| Nitric oxide synthase 3 | Nos3 | Mm00435217_m1 |
| NLR Family Pyrin Domain Containing 3 | Nlrp3 | Mm00840904_m1 |
| Nuclear receptor subfamily 3 group C member 1 | Nr3c1 | Mm00433832_m1 |
| Nuclear receptor subfamily 3 group C member 2 | Nr3c2 | Mm01241596_m1 |
| Placental growth factor | Plgf | Mm00435613_m1 |
| Serum/glucocorticoid regulated kinase 1 | Sgk1 | Mm00441380_m1 |
| Sodium channel epithelial 1 subunit alpha | Scnn1α | Mm00803386_m1 |
| Solute carrier family 9 (sodium/hydrogen exchanger), member 1 | Nhe1 | Mm00444270_m1 |
| TIMP metallopeptidase inhibitor 1 | Timp1 | Mm01341361_m1 |
| Transforming growth factor beta 1 | Tgfβ1 | Mm01178820_m1 |
| Transforming growth factor beta 2 | Tgfβ2 | Mm00436955_m1 |
| Transforming growth factor beta 3 | Tgfβ3 | Mm00436960_m1 |
| Tumour necrosis factor | Tnf | Mm00443258_m1 |
| Vascular cell adhesion molecule-1 | Vcam-1 | Mm01320970_m1 |
| Vascular endothelial growth factor A | Vegfa | Mm00437306_m1 |
| Ubiquitin C (reference gene for placenta, heart and kidney) | Ubc | Mm01198158_m1 |
| β-actin (reference gene for mesenteric arteries) | Actb | Mm02619580_g1 |

assessed with the Aperio ImageScope (v12.3.0.5056) software. The total blood space (including the combined fetal and maternal blood space area), and the total cross-sectional area of the junctional zone and the labyrinth zone was measured by a blinded assessor (102).

## Statistical analysis

Data were assessed for normal (Gaussian) distribution and the differences between the two groups at each time point were statistically tested either nonparametrically with a Mann–Whitney test or parametrically with an unpaired t test, as appropriate. Fetal and placental size were assessed using a linear mixed-effects model, with a fixed effect for treatment group and random effect for each litter. P-values were calculated for treatment effect using nested ANOVA. Myograph dose–response curves were produced using a nonlinear regression analysis (log[agonist] versus response–four parameters). Comparison of responses to the agonist were tested for significance using mixed-effects analysis, with Šidák correction for multiple comparisons. P-values < 0.05 were considered significantly different. Statistical analysis was performed using GraphPad Prism 8 software.

# Data Availability

The data that support the findings of this study are available from the corresponding author, upon reasonable request.

# Supplementary Information

# Acknowledgements

We would like to acknowledge the staff at the Austin Health BioResources Facility (Heidelberg) for assisting with the care of the animals. Support for this work was provided by the Trevor B Kilvington Bequest. Salary support was provided by the National Health and Medical Research Council Fellowships to NJ Hannan (#1146128). The funder played no role in study design or analysis.

## Author Contributions

N de Alwis: data curation, formal analysis, validation, investigation, visualization, methodology, and writing—original draft, review, and editing.
NK Binder: conceptualization, data curation, formal analysis, validation, methodology, project administration, and writing—original draft, review, and editing.
S Beard: data curation, formal analysis, investigation, and writing—review and editing.
YTM Mangwiro: conceptualization, data curation, formal analysis, investigation, methodology, project administration, and writing—original draft, review, and editing.
E Kadife: data curation, formal analysis, investigation, and writing—original draft, review, and editing.
JSM Cuffe: data curation, formal analysis, investigation, methodology, and writing—original draft, review, and editing.
E Keenan: formal analysis, methodology, and writing—review and editing.
BR Fato: data curation, formal analysis, investigation, and writing—review and editing.
TJ Kaitu'u-Lino: conceptualization, resources, supervision, and writing—review and editing.
FC Brownfoot: conceptualization, resources, supervision, and writing—review and editing.
SA Marshall: conceptualization, data curation, formal analysis, methodology, and writing—original draft, review, and editing.
NJ Hannan: conceptualization, resources, data curation, formal analysis, supervision, funding acquisition, project administration, and writing—original draft, review, and editing.

## Conflict of Interest Statement

The authors declare that they have no conflict of interest.

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
