## [Reviewer comments · Life Science Alliance]

Life Science Alliance

The L-NAME mouse model of preeclampsia and impact to long-term maternal cardiovascular health

Natasha de Alwis, Natalie Binder, Sally Beard, Yeukai Mangwiro, Elif Kadife, James Cuffe, Emerson Keenan, Bianca Fato, Tu'uhevaha Kaitu'u-Lino, Fiona Brownfoot, Sarah Marshall, and Natalie Hannan

DOI: <https://doi.org/10.26508/lsa.202201517>

Corresponding author(s): *Natasha de Alwis, University of Melbourne and Natalie Hannan, University of Melbourne*

Review Timeline:

Submission Date:	2022-05-04
Editorial Decision:	2022-05-30
Revision Received:	2022-06-23
Editorial Decision:	2022-07-18
Revision Received:	2022-07-19
Accepted:	2022-07-20

Scientific Editor: Novella Guidi

Transaction Report:

May 30, 2022

Re: Life Science Alliance manuscript #LSA-2022-01517-T

Dr. Natasha de Alwis
The University of Melbourne
Department of Obstetrics & Gynaecology
Level 4, Mercy Hospital for Women
163 Studley Road
Melbourne, VICTORIA 3084
Australia

Dear Dr. de Alwis,

Thank you for submitting your manuscript entitled "The L-NAME mouse model of preeclampsia and impact to long-term maternal cardiovascular health" to Life Science Alliance. The manuscript was assessed by expert reviewers, whose comments are appended to this letter. We invite you to submit a revised manuscript addressing the Reviewer comments.

Thank you for this interesting contribution to Life Science Alliance. We are looking forward to receiving your revised manuscript.

Sincerely,

B. MANUSCRIPT ORGANIZATION AND FORMATTING:

Reviewer #1 (Comments to the Authors (Required)):

1. This manuscript has investigated whether a common mouse model of pre-eclampsia, induced by inhibiting nitric oxide, can induce lasting changes in cardiovascular function in the mouse following recovery from pre-eclampsia. The paper demonstrates that the model induces hypertension in the absence of vascular dysfunction and that the effects are transient, where blood pressure is fully recovered by ~1-week post-pregnancy. There were really no lasting effects on the cardiovascular system other than a tendency for increased TNF α expression in hearts of pre-eclampsia mice.
2. For each main point of the paper, please indicate if the data are strongly supportive. If not, explicitly state the additional experiments essential to support the claims made and the timeframe that these would require.
 - a. L-NAME convincingly increased blood pressure in pregnant mice when delivered from day 7. However it is noted that there is a large spread of data with a lot of cross-over between groups (i.e. around a quarter of the control mice had higher mean blood pressure than L-NAME treated mice). With n>40 used in the L-NAME group in this study, it may not be a practical model for labs that are limited in resources- can you comment on the power / sample size required to undertake an experiment with significant blood pressure reduction by an intervention as a primary endpoint?
 - b. There were no differences in albumin:creatinine- can this please be presented like the other graphs with individual data to give the reader and idea of the data variation. The histology panels in figure 2 show potentially interesting changes in the kidneys, but it is not clear on how consistent this was across animals. Can quantification of this data be provided? Or images from multiple animals? Can the figure include arrows to the relevant points on the representative image to match description in figure legend? Does renal damage correlate with blood pressure?
 - c. For figure 3 E-F the total number of pups have been reported- when in fact pups should be averaged across a litter so that n is equivalent to the number of different mothers.
 - d. In figures 4-5 selected samples have been used for analysis of gene expression and circulating factors, but it is not clear how these samples were selected. Given the large degree of variation in blood pressures in the groups, it is important to show in a supplementary file the physiological parameters for the samples used in further analysis, and justification for why only a small subset of samples were used.
 - e. In the same vein as above, vessels from a sub-cohort of mice were assessed for vascular function ex vivo- can you please report on the physiological parameters for those particular individuals used.
 - f. A significant limitation of the model appears to be that ex vivo vascular function does not match with blood pressure measured in vivo- i.e. L-NAME has not caused a rightward shift in vasorelaxation. The authors have provided a nice discussion of this point, where they hypothesise that the effect of L-NAME is transient in the presence of circulating L-NAME and has effectively been 'washed out' in ex vivo resistance arteries. Does this indicate that L-NAME is not inducing any permanent changes during pregnancy? There are a few questions around vascular function- in that the response of controls appears to have a rightward shifted EC50 to what may be expected for acetylcholine responses- can the authors comment on this? The acetylcholine curve post-pregnancy is more what is expected with EC50 around 10⁻⁷, so it seems that the EC50 has shifted during pregnancy, is this expected?
 - g. For figure 6A the blood pressure drop in both groups post-pregnancy appears to be counter-intuitive. Generally there is a lowering of blood pressure in pregnancy - so can the authors explain why mean blood pressure drops down to only ~80 mmHg 10 weeks post-pregnancy? And could the data be extended to include matched mice during pregnancy showing the higher blood pressure in L-NAME vs. control mice, and the rate at which it returns to normal?
 - h. TNF α seems to be increased in hearts post-preeclampsia. This is an intriguing finding that deserves more exploration. Is it possible to look at other cytokines or inflammatory markers? Could the location of increased TNF α be determined using IHC?
3. Lastly, indicate any additional issues you feel should be addressed (text changes, data presentation, statistics etc.).
 - All bars graphs could be presented with individual points shown

Reviewer #2 (Comments to the Authors (Required)):

The current work by de Alwis and colleagues provides important preclinical data from an L-NAME mouse model of preeclampsia. The work shows that, despite the model inducing phenotypes akin to preeclampsia during pregnancy, this pathophysiology did not continue post-delivery. Overall, the manuscript reads well, and the data presented is informative and relevant to the field; however, there are some minor recommendations that should be considered and/or addressed.

Introduction:

The introduction reads well.

Methods:

1. What was the rationale for the concentration of L-NAME administered? Also, what was the rationale for the timing of L-NAME administration? Are these based on previously published work? In the introduction the authors state that "However, these murine studies using L-NAME differ in their protocols, using varied concentrations of L-NAME, routes of administration, timing of exposure, and murine breeds, that could respond differently to L-NAME [37]." This is an important point that appears to mostly be overlooked throughout the manuscript. Would the authors speculate that a higher L-NAME dose that may recapitulate a more severe disease phenotype result in prolonged cardiovascular dysfunction post-delivery? Given the simplicity of this model, it may benefit future studies to examine this to determine whether the severity of L-NAME-induced insult impacts maternal physiology after pregnancy. This consideration should be expanded on.
2. For the heart, was the left ventricle dissected and analysed separately from the other chambers or was total heart analysed? Some clarification for organ processing is needed.
3. Suggestion - a table may be an easier way to present the information for the qRT-PCR primers
4. Why were different reference genes used for different tissue? It is becoming increasingly common to normalise gene expression relative to the geometric mean of multiple reference genes rather than only one - could the authors provide comment on this?
5. Can the authors provide the sensitivity range for the ELISA kits used?

Results:

1. Figure 1 - specify that it is arterial blood pressure.
2. Can the authors include the statistical analysis used in each figure legend, as the authors specify that both parametric and non-parametric analyses were performed?
3. It appears that the post-delivery data has been split by each age category and analysed between study groups, but the way the data is graphically presented a two-way ANOVA would be the more appropriate choice of analysis. What was the rationale for the choice of statistical analysis?
4. The observed structural changes in the kidney is interesting, but some direction on the figure to point out these changes would be beneficial.
5. What is the rationale for expressing the Mmp data as a ratio to expression of inhibitor? Can the authors provide the expression of both Mmp and inhibitor separately and then as a ratio?
6. Did L-NAME administration induce brain sparing or just reduced birth weight?
7. In the discussion the authors state that "We did not observe significant changes in expression of inflammatory, or endothelial dysfunction-related genes in the mesenteric arteries post-delivery. However, there is limited data due to a low sample size." Is n=8/9 per group considered low?

Discussion:

The discussion is well written and comprehensive, albeit lengthy, which at times detracts from the importance of the findings. It is recommended that the discussion be edited down.

Reviewer #3 (Comments to the Authors (Required)):

Paper by de Alwis contains an impressive amount of work using a mouse model involving a vasoconstrictor, L-name to assess impacts on maternal blood pressure and vascular function in pregnancy and post-partum. The study also looks at changes in key organs like the placenta, as well as the kidney and heart of the mother. On a whole, the paper is very well written and conclusions substantiated. I only have minor comments and suggestions for improvement -mainly aimed at increasing the clarity of the data and interpretations. These are listed below

1. What is the rationale for the starting L-name treatment on e7.5? Is this equivalent to when maternal blood pressure may be expected to rise in PE women? Please include a sentence in the methods
2. Was sex of the fetuses/pups recorded?
3. Which placenta was taken from each litter? was this random or based on litter order/weight within the litter? Was there an equal representation of female and male placentas across the litters per group analysed for histological and molecular assays?
4. Was there any effect of the L-name treatment on gestational length?
5. Why were pups culled immediately after birth, rather than leaving the mums to support them through to weaning, which would be the normal situation? Include a sentence explaining the rationale in the methods.
6. Description of how pups and mums were culled post birth needs to be included.
7. Include details of the inter and intraassay CVs for the ELISAs/kits used to measuring maternal circulating factors.
8. Did the reference genes used in qPCR change per group? Please include a comment about this in the methods
9. For kidney histology were the three sections analysed per condition from different animals? What parameters were assessed in the kidney sections? Was it the medulla or cortex that was investigated? Please clarify in the text
10. How many litters per group were analysed for placental histology? What is meant by placental blood space volume? Is that

fetal and maternal blood spaces volume combined? Were histological analyses conducted blind to the group? Please clarify in the text and include an image of placental cross-section to indicate the compartments measured

11. Please include details of the statistical test/s used for each data shown, in the figure legends. All graphs should show individual points (there are a few which are instead bar charts and this does not allow an appreciation of the values for all the samples per group). All graphs should start at zero or show a break in the axis (eg this isnt the case for Fig 3b)

12. Include annotations on figure 2 to show the region/compartments of the kidney analysed that help to link better to the descriptions of changes in the text

13. Each conceptus/pup is a repeat of the mother and thus all data and any statistical analyses in figure 3 should involve the average value per litter or a mixed model approach to account for this (so sample size is litter/mum not the individual fetus/pup - which inflates the sample size)

14. Figure 4, how many placentas per litter were analysed? Include details in the legend

15. Please show individual values for Mmp9 and Timp for the maternal postpartum tissues in the supplementary data (so it is clear whether a change in both or one is driving the altered ratio between them).

16. The finding of altered vascular reactivity in L-name exposed dams post-pregnancy is subtle but interesting. I realise it would be out of scope for this study, but further work assessing the impact of a superimposed hit like a high salt diet or in animals as they get older would be valuable to see if changes in vascular reactivity may be exacerbated. Perhaps this could be mentioned in the discussion to help direct future work?

17. Given the changes in Hmox1 expression, it would be valuable if the authors measured oxidative stress levels, such as by oxyblot or MDA staining to see if the placentas may be stressed in this l-name model. Similarly it would be interesting to measure levels of ROS in the maternal blood. If L-name treated mice are not exhibiting oxidative stress, it could be the explanation for the subtle effects on maternal postpartum health

18. Given the change in Mmp9:timp, it would also be valuable if the authors assessed if there were any persistence of pathological changes in the maternal kidney postpartum, including leukocyte influx in the l-name treated dams.

RESPONSE TO REVIEWERS COMMENTS

Reviewer #1:

1. This manuscript has investigated whether a common mouse model of pre-eclampsia, induced by inhibiting nitric oxide, can induce lasting changes in cardiovascular function in the mouse following recovery from pre-eclampsia. The paper demonstrates that the model induces hypertension in the absence of vascular dysfunction and that the effects are transient, where blood pressure is fully recovered by ~1-week post-pregnancy. There were really no lasting effects on the cardiovascular system other than a tendency for increased TNFalpha expression in hearts of pre-eclampsia mice.

RESPONSE: We thank the reviewers for generously taking their time to review our manuscript. We have responded to each comment below.

- a. L-NAME convincingly increased blood pressure in pregnant mice when delivered from day 7. However it is noted that there is a large spread of data with a lot of cross-over between groups (i.e. around a quarter of the control mice had higher mean blood pressure than L-NAME treated mice). With $n > 40$ used in the L-NAME group in this study, it may not be a practical model for labs that are limited in resources- can you comment on the power / sample size required to undertake an experiment with significant blood pressure reduction by an intervention as a primary endpoint?

RESPONSE: We thank the reviewer for their question. A strength of this study was the large sample number in each group. We used these large numbers because we had several timepoints of interest post-pregnancy, and a subset of mice were euthanised at each timepoint for blood and other organ assessments. These large numbers ensured we would have ample numbers to assess blood pressure measurements at the final timepoint. Therefore, while it isn't necessary to run such larger numbers to achieve the convincing data in Figure 1, it was necessary to have these numbers to achieve sufficient power in the subsequent timepoints assessed.

We have added this to the methods see page 5, lines 114-115; which now reads:

“In order to have sufficient power for the final timepoints, a larger cohort of mice were required. Three week-old...”

Further, we have provided the specific mean and standard deviation values for the observed change in blood pressure in Figure 1 so other researchers can use this information for sample size calculations.

Please see page 10, lines 230-233 as below:

“L-NAME administration significantly increased mean arterial blood pressure at E14.5 (Control 103.7 ± 20.89 , L-NAME 126.5 ± 20.92 ; Mean \pm SD) ($p < 0.0001$; Figure 1A) and E17.5 (Control 111.9 ± 22.66 , L-NAME 130.1 ± 18.52 ; Mean \pm SD) ($p = 0.0001$; Figure 1B).”

- b. There were no differences in albumin:creatinine- can this please be presented like the other graphs with individual data to give the reader and idea of the data variation. The histology panels in figure 2 show potentially interesting changes in the kidneys, but it is not clear on how consistent this was across animals. Can quantification of this data be provided? Or images from multiple animals? Can the figure include arrows to the relevant points on the representative image to match description in figure legend? Does renal damage correlate with blood pressure?

RESPONSE: We thank the reviewer for their comments. Figure 2 has now been updated to present the individual points.

Due to tissue processing and staining issues, we were only able to analyse kidneys from two animals from the control and L-NAME groups. Arrows were used to point out the regions of interest (necrosis, inflammation, tubule changes) in the figure (*Figure 2*).

Due to the low number of samples available, descriptive analysis of these regions was more appropriate. We were unable to correlate kidney structural changes with blood pressure due to having too few samples for imaging, but this will be considered in future studies.

- c. For figure 3 E-F the total number of pups have been reported- when in fact pups should be averaged across a litter so that n is equivalent to the number of different mothers.

RESPONSE: We thank the reviewer for this suggestion. As a result of this suggestion, we sought advice from a biostatistician, and have now analysed the data using a linear mixed-effects model. This mixed-effects model takes into consideration the variability of the fetal/pup sizes within each litter using a random effect term. This method was suggested by Reviewer 3, point 13. The methods, figure and related results have been updated to reflect this change - *please see changes to statistical methods on page 9, lines 219-222, Figure 3, and results on page 10-11, lines 256-262*. Our biostatistician has also been added as a co-author of this manuscript.

- d. In figures 4-5 selected samples have been used for analysis of gene expression and circulating factors, but it is not clear how these samples were selected. Given the large degree of variation in blood pressures in the groups, it is important to show in a supplementary file the physiological parameters for the samples used in further analysis, and justification for why only a small subset of samples were used.

RESPONSE: We thank the reviewer for this suggestion.

In Figure 4, we assessed gene expression in placentas collected from the subset of mice that were culled at E17.5 of pregnancy. We were unable to collect the placentas from the mice that gave birth to their pups, as the dams consumed the placentas. We chose 1-3 placentas from each dam at random. Samples with low RNA yield at extraction were excluded. We have updated Figure 4 legend to include these details.

Please see figure 4 legend:

“Figure 4. Effect of L-NAME administration on placental gene expression at E17.5. [...] Control n=6 mice, L-NAME n=10 mice, 1-3 placentas were chosen at random from each. Samples with low RNA yield at extraction were excluded.”

Similarly in Figure 5A-C, the ELISA data presents the levels of the circulating factors in the subset of mice that were culled at E17.5, as we collected blood through cardiac puncture under anaesthesia at cull.

In Figure 5D and E, again we performed myograph experiments on vessels collected from all dams in the subset that were culled at E17.5. The dissected vessels were assessed for smooth muscle and endothelial integrity before they were included in the experiment. Due to technical difficulties in mounting very small arteries on the myograph, damage meant that some arteries

could not be used. This occurred at random due to the technique, and not due to any selection bias. We have clarified this in the methods as below.

Please see page 6, lines 168-169:

“Confirmation of response to KPSS, and greater than 80% relaxation was required for inclusion of the vessel.”

The fetal and placental parameters corresponding to this subset of mice culled at E17.5 are presented in Figure 3A-D. As suggested, we have now also added a graph of blood pressure data corresponding to this E17.5 cohort into the Supplementary Data. *Please see Supplementary Figure S4.*

- e. In the same vein as above, vessels from a sub-cohort of mice were assessed for vascular function ex vivo- can you please report on the physiological parameters for those particular individuals used.

RESPONSE: At each time point, a subset of mice were culled for collection of blood, organs and blood vessels. Mice were randomly allocated for cull at each time point. As described above (point d), mesenteric arteries from all dams were assessed at each time point, but as is an important feature of validation of myograph practice, only those that had intact smooth muscle and endothelial layers were included in the analysis. This is important to confirm that the vessel is not damaged during the mounting process, and that the vessel is still functionally active and able to respond to stimuli.

- f. A significant limitation of the model appears to be that ex vivo vascular function does not match with blood pressure measured in vivo- i.e. L-NAME has not caused a rightward shift in vasorelaxation. The authors have provided a nice discussion of this point, where they hypothesise that the effect of L-NAME is transient in the presence of circulating L-NAME and has effectively been 'washed out' in ex vivo resistance arteries. Does this indicate that L-NAME is not inducing any permanent changes during pregnancy? There are a few questions around vascular function- in that the response of controls appears to have a rightward shifted EC50 to what may be expected for acetylcholine responses- can the authors comment on this? The acetylcholine curve post-pregnancy is more what is expected with EC50 around 10⁻⁷, so it seems that the EC50 has shifted during pregnancy, is this expected?

RESPONSE: We thank their reviewer for their questions. Regarding the effect of L-NAME on the vasculature, we did initially speculate, as mentioned, that L-NAME does not seem to have a permanent effect on the vasculature, as when it is washed out at E17.5, the vessels do not have altered vascular reactivity. However, because there are changes in vascular reactivity and signs of inflammation in the heart and kidney at 10 weeks postpartum, it suggests that L-NAME did have some effect – that possibly the effects induced by L-NAME in pregnancy become worse over time. We also have noted in our discussion that we have only looked at phenylephrine and acetylcholine pathways in vascular function – chosen because they are most often used for mesenteric artery myograph studies. However, L-NAME may have altered other pathways involved in the vascular response that we did not assess – this is something we are interested in examining in the future, especially because we noted that circulating ET-1, sFLT-1 and CRP were altered. However, it was beyond the scope of this paper.

We have detailed this as below – please see page 17, lines 471-474 and 478-479:

“This would imply the administration of L-NAME may have exerted some permanent effects, either directly or indirectly (as the hypertensive phenotype may have driven other indirect changes that required time to become apparent), and further studies would be required to examine this. [...] Assessing vascular responses to other agonists may provide additional insight into other pathways that may be altered by L-NAME.”

Regarding the shift in EC50 in the pregnant vessels – in this paper we did not evaluate the differences across different time points, but rather between the L-NAME and control groups within each time point. The mice in pregnancy were injected with either the vehicle or treatment prior to cull, which may have induced a stressed, more vasoconstrictory state, compared to post-pregnancy where they were not administered any treatment before cull. It is known that pregnant and non-pregnant vessels respond differently, but because we did not control perfectly to assess changes over time, or assess pre-pregnancy vascular reactivity, we do not think we can speculate on this finding here.

- g. For figure 6A the blood pressure drop in both groups post-pregnancy appears to be counter-intuitive. Generally there is a lowering of blood pressure in pregnancy - so can the authors explain why mean blood pressure drops down to only ~80 mmHg 10 weeks post-pregnancy? And could the data be extended to include matched mice during pregnancy showing the higher blood pressure in L-NAME vs. control mice, and the rate at which it returns to normal?

RESPONSE: We thank the review for their comments and suggestions. We suggest that the maternal blood pressure (in both the control and L-NAME groups) post-partum reduction as mice aged, was largely due to the mice acclimatizing to the process of having their tail cuff blood pressure measurements performed (over the 13 weeks (pregnancy+post-partum)). Of important note, in this paper, we did not assess changes in the parameters from pre-pregnancy through pregnancy, and then post-partum. Rather, simply between the L-NAME (preeclampsia model) and control groups at each time point. Assessing changes over time and showing matched data in pregnancy and post-delivery warrants investigation in a focused paper in the future, but unfortunately was beyond the scope of the current study.

h. TNFalpha seems to be increased in hearts post-preeclampsia. This is an intriguing finding that deserves more exploration. Is it possible to look at other cytokines or inflammatory markers? Could the location of increased TNFa be determined using IHC?

RESPONSE: We thank the reviewer for this suggestion. In Supplementary Figure S11, we present the expression of a number of genes in the hearts, including other genes involved in inflammation, including interleukin-6 (Il-6; Supp Figure S11E), inflammasome gene Nlrp3 (Supp Figure S11K) and Tgf β 1, 2 and 3 (Supp Figure S11O-Q) which are involved in immune regulation. These genes were not altered at 10 weeks post-delivery. All hearts post-delivery were collected for RNA extraction and assessment of gene expression. Hence, we do not have the samples for IHC, but we agree this would be interesting for future studies.

We have stated the above in our discussion, page 17, lines 464-467:

“Further studies to determine if there is an increase in Tnf protein production and other markers of inflammation are needed to confirm this. Exploration of the cardiac structure and function may be helpful in evaluating whether our model produces the same left ventricular dysfunction seen post-preeclampsia.”

3. All bars graphs could be presented with individual points shown

RESPONSE: We thank the reviewer for their suggestion. We have altered Figure 2, 6 and 7 in the main manuscript figures to show all data points.

Reviewer #2:

The current work by de Alwis and colleagues provides important preclinical data from an L-NAME mouse model of preeclampsia. The work shows that, despite the model inducing phenotypes akin to preeclampsia during pregnancy, this pathophysiology did not continue post-delivery. Overall, the manuscript reads well, and the data presented is informative and relevant to the field; however, there are some minor recommendations that should be considered and/or addressed.

Introduction:

The introduction reads well.

RESPONSE: We thank the reviewer for the feedback on our manuscript, and have made changes as suggested - detailed below.

Methods:

1. What was the rationale for the concentration of L-NAME administered? Also, what was the rationale for the timing of L-NAME administration? Are these based on previously published work? In the introduction the authors state that "However, these murine studies using L-NAME differ in their protocols, using varied concentrations of L-NAME, routes of administration, timing of exposure, and murine breeds, that could respond differently to L-NAME [37]." This is an important point that appears to mostly be overlooked throughout the manuscript. Would the authors speculate that a higher L-NAME dose that may recapitulate a more severe disease phenotype result in prolonged cardiovascular dysfunction post-delivery? Given the simplicity of this model, it may benefit future studies to examine this to determine whether the severity of L-NAME-induced insult impacts maternal physiology after pregnancy. This consideration should be expanded on.

RESPONSE: The concentration of L-NAME used in this study was based on previous studies in rats, and pilot studies in our laboratory that determined an effective dose to increase blood pressure. In line with this, we have added references to others studies that use the same doses.

Please see methods, page 5, line 127:

"L-NAME (50mg/kg/day; Sigma-Aldrich, St. Louis, MO, USA [30, 46])..."

[30] C. Motta, C. Grosso, C. Zanuzzi, D. Molinero, N. Picco, R. Bellingeri, F. Alustiza, C. Barbeito, A. Vivas, M. Romanini, Effect of Sildenafil on Pre-Eclampsia-Like Mouse Model Induced By L-Name, *Reproduction in Domestic Animals* 50(4) (2015) 611-616.

[46] A.M. ShamsEldeen, M.N. Mehesen, B.E. Aboulhoda, L.A. Rashed, M.M. Elsebaie, E.A. Mohamed, M.M. Gamal, Prenatal intake of omega-3 promotes Wnt/ β -catenin signaling pathway, and preserves integrity of the blood–brain barrier in preeclamptic rats, *Physiological Reports* 9(12) (2021) e14925.

L-NAME was administered from E7.5 to E17.5 of pregnancy, to model the human equivalent gestation of end of first trimester/early second trimester to term. This was to try to model early-onset preeclampsia, where we see signs of hypertension prior to term – which is what we found with the elevation of blood pressure even at E14.5 of pregnancy. We did not begin administration earlier to avoid causing fetal loss, but found that we were still able to alter fetal/placental growth with this timing.

Please see the following statements in the manuscript that reflect this. Page 5, lines 129-131:

“...was administered daily from embryonic day (E)7.5 to E17.5 of pregnancy (approximately early second trimester to term in human equivalent to model early onset disease) via 100 μ L subcutaneous injection. ...”

It is possible that a higher dose of L-NAME could recapitulate a more severe long-term phenotype. Some have used lower and higher L-NAME doses, we used a medium dose range and examined various doses in the laboratory prior to proceeding. In addition, preeclampsia is a complex condition that features the impairment of many different pathways. Using L-NAME in this study, we aimed to inhibit the nitric oxide pathway to increase blood pressure and impair fetal growth. Following this study, rather than increasing this dose, we hypothesise impairing another pathway simultaneously may be more physiologically relevant. Our future studies will focus on a second hit model – potentially with a high fat/salt/sugar diet. Furthermore, our mice strain did not have a cardiovascular disease risk phenotype at the outset, which is a key consideration in long-term models – thus using mice with predisposition to cardiovascular disease may be more valuable than simply increasing the dose of L-NAME, and be more closely mimicking what we think is happening in humans.

Please see page 18, lines 502-507, where we have highlighted future studies:

“Assessment using a strain that that is more sensitive to L-NAME, may demonstrate a more persistent chronic effect on the cardiovascular system [37]. Additionally, it would be interesting to use a strain or genetic model with increased cardiovascular disease risk, potentially an obese or high salt diet mouse model, or a strain genetically modified to enhance vascular dysfunction, mimicking predisposition to preeclampsia and long-term cardiovascular disease.”

2. For the heart, was the left ventricle dissected and analysed separately from the other chambers or was total heart analysed? Some clarification for organ processing is needed.

RESPONSE: We thank the reviewer for their suggestion. Whole hearts were analysed. We have clarified this in the methods, *page 6, lines 175-176 as below.*

“RNA was extracted from the RNAlater preserved placenta, heart (whole), kidney, and mesenteric arteries...”

3. Suggestion - a table may be an easier way to present the information for the qRT-PCR primers

RESPONSE: We thank the reviewer for their suggestion. We have presented the primer information in a table for ease. *Please see Table 1 on pages 7-8.*

4. Why were different reference genes used for different tissue? It is becoming increasingly common to normalise gene expression relative to the geometric mean of multiple reference genes rather than only one - could the authors provide comment on this?

RESPONSE: We thank the reviewer for highlighting this. Our group has previously validated the appropriate reference genes that are stable in the each tissue assessed. Further, in analysis we ensured that the reference genes were stable in our study within the L-NAME and control groups.

We have added the following to clarify see. Please see page 7, lines 185-187:

“All expression data were normalized to expression of reference gene (chosen based on their stability in each tissue) as an internal control and calibrated against the average Ct of the control samples, with each biological sample run in technical duplicate.”

5. Can the authors provide the sensitivity range for the ELISA kits used?

RESPONSE: We thank the reviewer for this suggestion. We have now added the ELISA kit sensitivity where available into the methods as below.

Please see page 8, lines 193-195:

“...Mouse Endothelin-1 Quantikine ELISA Kit (sensitivity 0.207 pg/mL) and Mouse C-Reactive Protein/CRP Quantikine ELISA Kit (sensitivity 0.015 ng/mL)...”

Please see page 8, lines 199-200:

“...using the Mouse Albumin ELISA kit (sensitivity 5 ng/mL)...”

Results:

1. Figure 1 - specify that it is arterial blood pressure.

RESPONSE: We thank the reviewer for this comment. We have specified that we have measured mean arterial blood pressure in the figure legends as below.

“Figure 1. Effect of L-NAME administration on mean arterial blood pressure E14.5 and E17.5 of pregnancy.”

“Figure 6. Effect of L-NAME administration in pregnancy on A) mean arterial blood pressure...”

“Supplementary Figure S1. Effect of L-NAME administration on systolic and diastolic arterial blood pressure at E14.5 (A, B) and E17.5 (C, D) of pregnancy.”

“Supplementary Figure S8. Effect of L-NAME administration in pregnancy on A) systolic and B) diastolic arterial blood pressure post-delivery.”

2. Can the authors include the statistical analysis used in each figure legend, as the authors specify that both parametric and non-parametric analyses were performed?

RESPONSE: We thank the reviewer for their suggestion. We have added the details of statistical analysis into each figure legend.

3. It appears that the post-delivery data has been split by each age category and analysed between study groups, but the way the data is graphically presented a two-way ANOVA would be

the more appropriate choice of analysis. What was the rationale for the choice of statistical analysis?

RESPONSE: We thank the reviewer for highlighting this. In this study, we only evaluated differences between the L-NAME and control group within each timepoint. We did not analyse differences across time/across timepoints. Hence, it was more appropriate to do a t-test or Mann-Whitney test between the two groups at each timepoint. We have now updated the figure legends to include the statistical analysis used, and clarified that these tests were performed between the L-NAME and control groups at each time point.

4. The observed structural changes in the kidney is interesting, but some direction on the figure to point out these changes would be beneficial.

RESPONSE: We thank the reviewer for the suggestion. We have included arrows to highlight the regions of interest.

Please refer to Figure 2 and figure legend as below:

“Pathological features of L-NAME treated mice show inflammatory cell infiltration around regions of necrosis (black arrows) with haemoglobin (dark pink in section) and hyaline (light pink) casts that are surrounded by flattened nuclei and irregular cells (white arrows) (E-F).”

5. What is the rationale for expressing the Mmp data as a ratio to expression of inhibitor? Can the authors provide the expression of both Mmp and inhibitor separately and then as a ratio?

RESPONSE: We thank the reviewer for their question. We expressed the Mmp data as a ratio to its Timp inhibitor as it gives us an insight into whether a change in Mmp levels may actually bring about a functional change if altered. As suggested by the reviewer, we have presented the expression of the MMP genes and TIMP alone, before presenting them as a ratio. *Please see Supplementary Figures S10F-H and S11F-I.*

6. Did L-NAME administration induce brain sparing or just reduced birth weight?

RESPONSE: In this study we found that L-NAME reduced birthweight and fetal crown to rump length. We do not have data measuring fetal or pup head/brain size or weight. Hence, we cannot

rule out that there was brain sparing. In future studies we will be investigating the effect of L-NAME and drug treatments on both head size, and examination of the fetal brain morphology with the help of collaborators who are experts in the field. But this is unfortunately beyond the scope of this manuscript.

We have added the following as response to this – please see page 14, lines 369-370:

“Further studies are required to uncover the mechanisms behind this growth impairment, including whether there are any clinical parameters of interest including fetal brain sparing.”

7. In the discussion the authors state that "We did not observe significant changes in expression of inflammatory, or endothelial dysfunction-related genes in the mesenteric arteries post-delivery. However, there is limited data due to a low sample size." Is n=8/9 per group considered low?

RESPONSE: We thank the reviewer for pointing this out. In this particular case, we considered the sample size small because we had to pool the 8-9 samples we had in each group. This would have decreased the effective sample size and hence, our statistical power.

We have altered the discussion for clarity – please see page 17, lines 481-482:

“However, there is limited data as the samples had to be pooled, leading to a low effective sample size.”

Discussion:

The discussion is well written and comprehensive, albeit lengthy, which at times detracts from the importance of the findings. It is recommended that the discussion be edited down.

RESPONSE. We thank the reviewer for this suggestion. We have cut down on the discussion as much as possible, removing details covered in the results without losing the important context and discussion. Please see revised manuscript.

Reviewer #3:

Paper by de Alwis contains an impressive amount of work using a mouse model involving a vasoconstrictor, L-name to assess impacts on maternal blood pressure and vascular function in pregnancy and post-partum. The study also looks at changes in key organs like the placenta, as well as the kidney and heart of the mother. On a whole, the paper is very well written and conclusions substantiated. I only have minor comments and suggestions for improvement -mainly aimed at increasing the clarity of the data and interpretations. These are listed below.

RESPONSE: We thank the reviewer for their time in reading our manuscript, and have incorporated their suggestions as below.

1. What is the rationale for the starting L-name treatment on e7.5? Is this equivalent to when maternal blood pressure may be expected to rise in PE women? Please include a sentence in the methods

RESPONSE: We thank the reviewer for their comment. We began L-NAME administration on E7.5, which is the approximately the human equivalent of end of first trimester/early second trimester. Preeclampsia is defined as new onset hypertension after 20 weeks gestation. By starting administration at E7.5, we are able to induce hypertension by D14.5 which is approximately mid-way through gestation, modelling the clinical condition.

The methods have been updated to address this – see *page 5, line 129-131 as below.*

“...was administered daily from embryonic day (E)7.5 to E17.5 of pregnancy (approximately early second trimester to term in human equivalent to model early onset disease) via 100µL subcutaneous injection.”

2. Was sex of the fetuses/pups recorded?

RESPONSE: We thank the reviewer for this question. We did not record sex of the fetuses/pups in this study. However, this is a good suggestion, and will consider examining differences with fetal sex in future studies.

3. Which placenta was taken from each litter? was this random or based on litter order/weight within the litter? Was there an equal representation of female and male placentas across the litters per group analysed for histological and molecular assays?

RESPONSE: In this particular study, we did not look at the effect of fetal sex. We chose 1-3 placentas at random for analyses. These placentas were collected on from the subset of mice culled at E17.5, as the dams that gave birth ate their placentas. We have now specified this in the figure legends.

We have updated the Figure 4 legend (placenta gene expression) – please see page VII of the figures, as below:

“Control n=6 mice, L-NAME n=10 mice, 1-3 placentas were chosen at random from each.”

We have also updated Supplementary Figure S3 legend (placental histology):

“Placentas from the mice culled at E17.5 were chosen at random for analysis, each placenta from a different dam.”

4. Was there any effect of the L-name treatment on gestational length?

RESPONSE: We thank the reviewer for this question. We were also curious to know whether L-NAME may have induced preterm birth. However, there was no significant effect on gestational length. All mice gave birth on E19 (night), and thus all pups were culled on E19.5, the following morning.

We have added the following to state this – please see page 11, lines 263-264:

“L-NAME administration did not alter gestational length - all mice gave birth by the morning of E19.5.”

5. Why were pups culled immediately after birth, rather than leaving the mums to support them through to weaning, which would be the normal situation? Include a sentence explaining the rationale in the methods.

RESPONSE: We thank the reviewer for this suggestion. In this study, due to the large number of dams, we did not have the capacity to keep all the pups (in this study there were 400+ pups in

total). We do know that this may have altered long-term cardiovascular risk, and other studies are investigating these effects. We have addressed the benefits of studying the effect of lactation in our discussion.

Please see pages 18, lines 509-512 as below:

“Another important aspect would be to assess whether lactation may have an effect on cardiovascular health post-hypertensive pregnancy, as studies have shown that lactation may reduce vasocontractility, enhance vasorelaxation and increase vessel distensibility [104].”

6. Description of how pups and mums were culled post birth needs to be included.

RESPONSE: We thank the reviewer for their suggestion. We have updated the methods to include this.

Please see page 6, lines 147-148 for details of pup euthanasia:

“Day (D)1 pups were counted, weighed, and crown-to-rump length recorded before euthanasia by decapitation.”

Dams that gave birth were culled at the postpartum time points in the same way as those that were pregnant.

We have edited the methods to refer to these details, and make this clearer. Please see page 6, lines 149-152:

“At cull (anaesthesia and cervical dislocation), cardiac puncture was performed and urine, maternal organs (heart and kidney collected in RNA later) and intestinal tract collected as described above.”

7. Include details of the inter and intraassay CVs for the ELISAs/kits used to measuring maternal circulating factors.

RESPONSE: We thank the reviewer for their question. As reviewer 2 requested (Q5), the sensitivity of the ELISA kits has been provided. Further, we have clarified the inter and intra assay coefficients for the purchased ELISA kits is under 10% as below.

Please see page 8, lines 195-196:

“(R&D Systems, Minneapolis, MN, USA) respectively, according to manufacturer’s instructions (inter and intra assay coefficients of variation under 10%).”

8. Did the reference genes used in qPCR change per group? Please include a comment about this in the methods

RESPONSE: As stated in response to reviewer 2 (Q4), we used reference genes that we determined were stable within each tissue, even with treatment. We have added information in methods to clarify this.

Please see page 7, lines 185-187:

“All expression data were normalized to expression of reference gene (chosen based on their stability in each tissue) as an internal control and calibrated against the average Ct of the control samples, with each biological sample run in technical duplicate.”

9. For kidney histology were the three sections analysed per condition from different animals? What parameters were assessed in the kidney sections? Was it the medulla or cortex that was investigated? Please clarify in the text

RESPONSE: We thank the reviewer for their comments. We analysed whole kidneys from 2 animals per cohort. The regions of interest represented in the figures were only observed within cortex. Tubule/glomeruli structures including size/shape of nuclei, presence of immune infiltration and haemoglobin (dark pink in section) and hyaline (light pink) casts suggestive of necrosis and damage were the parameters assessed.

Please see updated methods on page 9, lines 207-209:

“Kidney histology was visualized and captured using a Nikon Eclipse Ci microscope and camera at 100µm magnification (n=3 sections/condition, whole kidneys from two dams were analysed per group).”

We have clarified region of interest in Figure 2 legend:

“Histology images of the cortex of PBS treated control mice...”

10. How many litters per group were analysed for placental histology? What is meant by placental blood space volume? Is that fetal and maternal blood spaces volume combined? Were histological analyses conducted blind to the group? Please clarify in the text and include an image of placental cross-section to indicate the compartments measured

RESPONSE: We thank the reviewer for their question. We were only able to collect placentas from the subset of mice that were culled at E17.5, as the dams that were left to give birth consumed the placentas. The placental blood space volume that we've presented in this study included the combined fetal and maternal blood spaces, and was assessed blind to sample group allocation. We have altered the methods to clarify this, and have also added a citation to a paper in which we have previously published this type of analysis, including an image of placental cross-section.

Please see Supplementary Figure S3 legend, clarifying number of samples as below:

“Placentas from the mice culled at E17.5 were chosen at random for analysis, each placenta from a different dam. Data presented as mean \pm SEM. n=3-5 mice/group.”

Please see clarification of histological assessments - page 9, lines 211-214 as below:

“General histological appearance was assessed with the Aperio ImageScope (v12.3.0.5056) software. The total blood space (including the combined fetal and maternal blood space area), and the total cross-sectional area of the junctional zone and the labyrinth zone was measured by a blinded assessor [47].”

[47] S.L. Helman, S.J. Wilkins, D.R. McKeating, A.V. Perkins, P.E. Whibley, J.S.M. Cuffe, D.G. Simmons, B.K. Fuqua, C.D. Vulpe, D.F. Wallace, J.L. O'Callaghan, E.S. Pelzer, G.J. Anderson, D.M. Frazer, The Placental Ferroxidase Zyklopen Is Not Essential for Iron Transport to the Fetus in Mice, *The Journal of Nutrition* 151(9) (2021) 2541-2550.

11. Please include details of the statistical test/s used for each data shown, in the figure legends. All graphs should show individual points (there are a few which are instead bar charts and this does not allow an appreciation of the values for all the samples per group). All graphs should start at zero or show a break in the axis (eg this isnt the case for Fig 3b)

RESPONSE: We thank the reviewer for their suggestions. We have now added the statistical tests used in each figure legend (as also requested by reviewer 2). We have also changed all bar graphs in the main figures to present individual points. The y-axis of Figure 3B has been edited to start from zero.

12. Include annotations on figure 2 to show the region/compartments of the kidney analysed that help to link better to the descriptions of changes in the text

RESPONSE: We thank the reviewer for the suggestion. As suggested by all reviewers, we have included coloured arrows to highlight the regions of interest.

Please refer to Figure 2, and figure legend as below:

“Pathological features of L-NAME treated mice show inflammatory cell infiltration around regions of necrosis (black arrows) with haemoglobin (dark pink in section) and hyaline (light pink) casts that are surrounded by flattened nuclei and irregular cells (white arrows) (E-F).”

13. Each conceptus/pup is a repeat of the mother and thus all data and any statistical analyses in figure 3 should involve the average value per litter or a mixed model approach to account for this (so sample size if litter/mum not the individual fetus/pup - which inflates the sample size)

RESPONSE: We thank the reviewer for this suggestion. As written in response to Reviewer 1 (1c), with advice with our biostatistician we have now analysed the data using a mixed effects model to account for differences between litters.

Please see updated Figure 3, and updated description of statistical analysis on page 9, lines 219-222.

14. Figure 4, how many placentas per litter were analysed? Include details in the legend

RESPONSE: We thank the reviewer for their question. As stated above in response 3, we were only able to collect placentas from the subset of mice that were culled at E17.5, as the dams that were left to give birth consumed the placentas. Of this subset, we chose 1-3 placentas per litter at random for assessment of gene expression.

We have updated the Figure 4 legend for the placental gene expression – changes as below:

“Control n=6 mice, L-NAME n=10 mice, 1-3 placentas were chosen at random from each.”

15. Please show individual values for Mmp9 and Timp for the maternal postpartum tissues in the supplementary data (so it is clear whether a change in both or one is driving the altered ratio between them).

RESPONSE: We thank the reviewer for this suggestion. As Reviewer 2 (Q5) also suggested, we have added the Mmp and Timp data alone, before presenting the ratio of their expression.

Please see Supplementary Figures S10F-H and S11F-I.

16. The finding of altered vascular reactivity in L-name exposed dams post-pregnancy is subtle but interesting. I realise it would be out of scope for this study, but further work assessing the impact of a superimposed hit like a high salt diet or in animals as they get older would be valuable to see if changes in vascular reactivity may be exacerbated. Perhaps this could be mentioned in the discussion to help direct future work?

RESPONSE: We agree with the reviewer that this would indeed be interesting future direction. In our discussion, we have highlighted that second hit with a high fat/salt/sugar diet, and using animals with a cardiovascular disease predisposition could be of interest to establish a model of long-term cardiovascular disease following preeclampsia.

Please see page 18, lines 504-507, where we have highlighted this:

“Additionally, it would be interesting to use a strain or genetic model with increased cardiovascular disease risk, potentially an obese or high salt diet mouse model, or a strain genetically modified to enhance vascular dysfunction, mimicking predisposition to preeclampsia and long-term cardiovascular disease.”

17. Given the changes in Hmox1 expression, it would be valuable if the authors measured oxidative stress levels, such as by oxyblot or MDA staining to see if the placentas may be stressed in this l-name model. Similarly it would be interesting to measure levels of ROS in the maternal

blood. If L-name treated mice are not exhibiting oxidative stress, it could be the explanation for the subtle effects on maternal postpartum health

RESPONSE: We thank the reviewer for this suggestion. This is indeed an interesting thought, and would be interesting to assess oxidative stress further in future studies. However, as detailed in response to Reviewer 2 (Methods – response 1) our next steps would be more focused on using a second hit to impair other pathways in addition to the nitric oxide pathway, or to use L-NAME an animal with a cardiovascular disease risk background, to more accurately model the complexity of preeclampsia (*as specified on page 18, lines 502-512*).

18. Given the change in Mmp9: timp, it would also be valuable if the authors assessed if there were any persistence of pathological changes in the maternal kidney postpartum, including leukocyte influx in the l-name treated dams.

RESPONSE: We thank the reviewer for this suggestion. We did look at the kidneys at each time point post-pregnancy. However, there were no persistent changes were observed in the 10wk post-partum kidney sections, and so this was not presented in this study.

July 18, 2022

RE: Life Science Alliance Manuscript #LSA-2022-01517-TR

Dr. Natasha de Alwis
University of Melbourne
Department of Obstetrics & Gynaecology
Level 4, Mercy Hospital for Women
163 Studley Road
Melbourne, VICTORIA 3084
Australia

Dear Dr. de Alwis,

Thank you for submitting your revised manuscript entitled "The L-NAME mouse model of preeclampsia and impact to long-term maternal cardiovascular health". We would be happy to publish your paper in Life Science Alliance pending final revisions necessary to meet our formatting guidelines.

- please add ORCID ID for secondary corresponding author. You should have received instructions on how to do so
- please upload your main and supplementary figures as single files and add a separate section for your figure legends to the main manuscript
- please add your author contributions to the main manuscript text
- please consult our manuscript guidelines <https://www.life-science-alliance.org/manuscript-prep> and make sure that your manuscript sections are in the correct order
- please use the format of 10 authors et al. in the references
- we encourage you to introduce the panels in your Figure legends in alphabetical order

Figure Check:

- please detail the panels more in legend of Figure S2, Figure S7, Figure S10, Figure S11, Figure S15

A. FINAL FILES:

B. MANUSCRIPT ORGANIZATION AND FORMATTING:

Sincerely,

Reviewer #1 (Comments to the Authors (Required)):

The authors have provided an extensive response to reviewers and have satisfactorily addressed all the comments. In fact residual concerns to point 1a, were addressed in the response to point 1b- whereby the smaller subset of mice used in further analysis had significantly increased blood pressure with L-NAME, which suggests that the model can be used with less mice required, making it a useful model for early onset pre-eclampsia.

Reviewer #2 (Comments to the Authors (Required)):

The revised version addresses all comments raised in the previous version, and any additional concerns were discussed appropriately in the rebuttal. No further revision needed.

Reviewer #3 (Comments to the Authors (Required)):

I am happy that the authors have taken on board my suggested changes. I believe these have improved the paper. I have no further suggested changes.

July 20, 2022

RE: Life Science Alliance Manuscript #LSA-2022-01517-TRR

Dr. Natasha de Alwis
University of Melbourne
Department of Obstetrics & Gynaecology
Level 4, Mercy Hospital for Women
163 Studley Road
Melbourne, VICTORIA 3084
Australia

Dear Dr. de Alwis,

Thank you for submitting your Research Article entitled "The L-NAME mouse model of preeclampsia and impact to long-term maternal cardiovascular health". It is a pleasure to let you know that your manuscript is now accepted for publication in Life Science Alliance. Congratulations on this interesting work.

DISTRIBUTION OF MATERIALS:

Again, congratulations on a very nice paper. I hope you found the review process to be constructive and are pleased with how the manuscript was handled editorially. We look forward to future exciting submissions from your lab.

Sincerely,
